

# Localized Coherence of Freak Waves

Arnida L. Latifah[1,2] and E. van Groesen[1,3]

[1]University of Twente, The Netherlands
[2]Indonesian Institute of Sciences, Bandung, Indonesia
[3]LabMath-Indonesia, Bandung, Indonesia

*Correspondence to:* Arnida L. Latifah (a.l.latifah@utwente.nl)

**Abstract.** This paper investigates in detail a possible mechanism of energy convergence leading to freak waves. We give examples of a freak wave as a (weak) pseudo-maximal wave to illustrate the importance of phase coherence. Given a time signal at a certain position, we identify parts of the time signal with successive high amplitudes, so-called group events, that may lead to a freak wave using wavelet transform analysis. The local coherence of the critical group event is measured by
its time spreading of the most energetic waves. Four types of signals have been investigated; dispersive focussing, normal sea condition, thunder storm condition, and an experimental irregular wave. In all cases presented in this paper, it is shown that a high correlation exists between the local coherence and the appearance of a freak wave. This makes it plausible that freak waves can be developed by local interactions of waves in a wave group and that the effect of waves that are not in the immediate vicinity is minimal. This indicates that a local coherence mechanism within a wave group can be one mechanism that leads to
the appearance of a freak wave.

## 1 Introduction

Understanding the mechanism of the freak wave phenomenon is intriguing for scientists, engineers, and mariners. The mechanisms that lead to freak waves are understandably diverse and it is not surprising that different freak waves exhibit different qualitative features (Liu and Mori, 2000). A review of the existing mechanisms of freak waves was presented by
Pelinovsky and Kharif (2008), Slunyaev et al. (2011).

We consider freak waves in unidirectional wave fields which satisfy the common definition of a freak wave, namely that the wave height exceeds approximately two times the significant wave height ($H_s$) or that the crest height exceeds $1.25 H_s$ (Kharif and Pelinovsky, 2006; Olagnon and van Iseghem, 2000; Dysthe et al., 2008; Kharif and Pelinovsky, 2003). Freak waves that are dominantly generated from wave energy convergence as a consequence of the random superposition
of many waves components with not necessarily strong nonlinearity is still under discussion (Wang et al., 2015; Onorato et al., 2013; Garret and Gemmrich, 2009; Gemmrich and Garrett, 2008; Slunyaev et al., 2005; Muller et al., 2005). Different from some papers (Haver, 2004; Kharif and Pelinovsky, 2003; Pelinovsky et al., 2000a), in which a freak wave is discussed as an accidental event from nowhere, that appears and disappears suddenly, we discuss freak waves in (mainly) random wave fields that exhibit long-life gradual growth and decay. Latifah and van Groesen (2012) described and predicted freak waves by mea-
suring the degree of phase coherence from a given time series at one position. It is the phase variance over an interval of the



dominant wave frequencies. In this paper, we investigate the local coherence computed from the local time spreading of the most energetic waves, which is determined by wavelet transform. Nowadays, wavelet transformation is widely applied to analyse freak waves (Hu et al., 2015; Kwon et al., 2015; Cherneva and Soares, 2014; Bai et al., 2015; Wang et al., 2015; Wu et al., 2010) as it has wider applicability than Fourier techniques (Lin and Liu, 2004).

In the study of Slunyaev et al. (2005), the calculation of the first derivative of the local group velocity in the time series shows the presence of regions of strong wave convergence or divergence near freak events where strong modulations occur. However, the question about the origin of the freak wave, whether it is naturally contained in the wave trains or induced by Benjamin Feir instability, is still open. Pelinovsky et al. (2011) discussed the scenario of a single freak wave in deep water by dispersive focussing of a unidirectional wave packet in linear theory and showed that the freak wave is originated from an

anomalous solitary wave. In addition, we also contribute in understanding the process and the origin of freak waves appearing in random wave fields.

In unidirectional linear waves, the focussing due to dispersion is one mechanism that causes a freak wave (Porubov et al., 2005; Slunyaev et al., 2005; Kharif et al., 2001; Brown and Jensen, 2001; Pelinovsky et al., 2000b; Baldock et al., 1996). If short waves with small group velocities are initially located in front of long waves having large group velocities, the long waves

will overtake the short waves with increasing time, and large-amplitude waves can appear. Afterwards, the long waves will be in front of the short waves, and the amplitude of the wave train will decrease (Kharif and Pelinovsky, 2003). This mechanism is observed in the type of dispersive focussing waves which are often used in hydrodynamic laboratories (Merkoune et al., 2013; Brown and Jensen, 2001; Clauss, 2002; Shemer et al., 2007, 2005; Grue et al., 2003). In random waves, this mechanism could also trigger a freak wave, but it is not as clear as in the dispersive focussing case. In the study of Wang et al. (2015),

they presented a freak wave in a random wave field that was generated from two successive wave groups with different main frequency and the higher frequency waves are in front of the other.

According to the study of Sergeeva et al. (2014) and Sergeeva and Slunyaev (2013), most of the long-living freak waves often occur on the background of intense wave groups. The evolution of modulated wave groups over large spatial and temporal scales were also a concern in the study of Viotti et al. (2013) and Grimshaw et al. (2001). Recently Cousins and Sapsis (2014,

2016) and Ruban (2013) underlined that the appearance of extreme events can be triggered by focussing energy in localized wave groups. Therefore, to identify the group profiles that can be the origin of freak waves appearance, they used envelope equations and identified the envelope of the dominant groups associated with the length scale and amplitude by a group detection algorithm. Further, they computed the probability of the group to develop an extreme event. The evolution of the freak waves is summarized into focussing-defocussing process of energy. During the generation, a single wave absorbs energy from

neighbouring waves, increases its amplitude, reaches a maximum and then returns its energy back to other waves (Xia et al., 2015; Slunyaev et al., 2005). According to Kharif et al. (2009), the transient change of the local energy of wave groups can be caught by wavelet analysis better than Fourier analysis.

In this paper we will consider the appearance of freak waves in evolving wave groups in space and time. The waves are generated from a signalling problem: at the influx position, say $x_0$, a given time signal $\eta(x_0, t)$ is influxed in one direction, the

positive $x$-axis. The resulting waves $\eta(x, t)$ may show a freak wave at certain time and space $(x_f, t_f)$ at which the amplitude is





larger than $1.25H_s$, which is taken as the definition of a freak wave in the rest of this paper. We will investigate this appearance by concentrating on successive high amplitudes in the initial signal, which will be called critical group events. We will apply the wavelet analysis for the identification of the energy spectral distribution in the group events.

This paper is organized into five sections started with this introduction. Section 2 starts with a motivation to investigate the
local coherence by showing the rapid decrease of the maximal amplitude when the coherence is decreased. Wavelet transformation is then described and shown to be better capable than Fourier methods to analyze the local phase of a wave. Section 3 starts with the selection of possible freak waves by estimating the critical group events from the influx signal that can lead to freak waves further downstream. The propagation of the most energetic group is then simulated to show the successive local energy convergence. We introduce quantitative measures of local coherence as one tool to predict the freak wave appear-
ance. Using numerical simulations of linear and nonlinear waves with the AB-equation (van Groesen and Andonowati, 2007; van Groesen et al., 2010), we compute the wave evolution and measure the local coherence of the time signal at several positions. We consider various wave types, a dispersive focussing wave and irregular waves, synthetic and experimental signals from MARIN hydrodynamic laboratory in Sect. 4. Conclusions are formulated in the final section.

## 2   Coherence and Wavelet Transform

In this section we will start to motivate and illustrate the role of coherence by considering maximal, pseudo-maximal (pm), and weak pseudo-maximal (wpm) signals that can describe freak waves. In Latifah and van Groesen (2012), the notion of a 'pseudo-maximal' signal was introduced for which the phases of all frequencies were band-limited. Below we also consider a less restrictive notion of weak pseudo-maximal signal, by restricting the phase only for the most energy carrying modes. The measure of phase coherence in these concepts uses Fourier transform that represents the energy and the phase as function of
the frequency. In Sect. 2.2, we describe the wavelet transform that is used in this paper to extract the local energy spectral distribution and the local phase as the time-frequency information of a given signal. Plots of the energy distribution over the frequencies will show that the wavelet transform improves results obtained with Fourier transform.

### 2.1   Signal coherence

Waves in the ocean at a specific position are described by a time signal. An irregular signal will have phases that are commonly
understood to be uniformly distributed in $(-\pi, \pi]$. Previous study (Latifah and van Groesen, 2012) defined maximal waves and pseudo-maximal waves. A maximal wave is a wave with all phases zero and has maximal amplitude equal to the integration of its two-sided absolute spectrum. Thus, we call a signal with all phases zero at some time (say $t = 0$) a maximal signal,

$$MS(t) = \int |\check{\eta}_0(\omega)| \cos(\omega t)\, d\omega.$$

At $t = 0$, all wave components contribute to a constructive interference, hence

$$MS(0) = \int |\check{\eta}_0(\omega)|\, d\omega.$$





This is the highest amplitude that is possible for given spectrum, $\breve{\eta}_0(\omega)$. In view of the assumption of uniform distribution of the phases, the chance for such a maximal wave vanishes.

A pseudo-maximal (pm) wave is a partly coherent wave, that is in between a completely irregular wave and a fully coherent maximal wave. For a given signal with random phase $\theta(\omega) \in (-\pi, \pi]$ as a function of wave frequencies with $\theta(\omega) = -\theta(-\omega)$, we consider a pm signal as the signal for which the phases are restricted for certain $\alpha \in (0,1)$ to the phases $\theta_\alpha(\omega) = \alpha\theta(\omega)$, as

$$[\eta_{pm}(t)]_\alpha = \int |\breve{\eta}_0(\omega)| \cos(\theta_\alpha(\omega) - \omega t)\, d\omega. \tag{1}$$

By taking a fraction $\alpha$ of the random phase, the maximal amplitude decreases and the background increases for increasing $\alpha$. For $\alpha = 0$, it is a maximal wave with coherent phases while for $\alpha = 1$ it is an irregular wave and the freak wave may disappear completely.

The phases of all frequencies in a pm signal are constrained as $|\theta(\omega)| < \alpha\pi$. We now define a weak pseudo-maximal (wpm) signal, $\eta_r(t)$, by restricting the phases of only the frequencies of large energy carrying modes (see Fig. 1 for an illustration). We also illustrate the importance of such restrictions for coherence by plotting the maximal, pm, and wpm signals of a given Jonswap spectrum in Fig. 2.

The restriction of wpm signal is typically for frequencies within one (or a half) standard deviation (sdv) around the mean frequency, $|\omega - \omega_m| < \sigma_\omega$ or less. Then we consider a signal, for $\alpha = [0,1]$ and define $\theta_\alpha$ as

$$\theta_\alpha(\omega) = \begin{cases} \alpha\theta(\omega), & \text{for } |\omega - \omega_m| < \sigma_\omega \\ \theta(\omega), & \text{elsewhere} \end{cases} \tag{2}$$

and get a signal that has maximal amplitude less than the maximal amplitude of the pm signal:

$$[\eta_r(t)]_\alpha = \int\limits_{-\infty}^{\infty} |\breve{\eta}(\omega)| \cos(\theta_\alpha(\omega) - \omega t)\, d\omega \leq [\eta_{pm}(0,0)]_\alpha. \tag{3}$$

In general the mean frequency is not necessarily equal to the peak frequency because the spectrum of waves, that is usually of Jonswap shape, is not symmetric around the peak frequency.

Figure 3 illustrates that the value of $\alpha$ significantly affects the maximal crest height and the wave evolution along the $x-$axis. The smaller the value of $\alpha$, the higher the value of the generated crest. On the other hand, variations in $\sigma_\omega$ influence much less the maximum elevation of the influx signal. In any case, the wave evolution is tremendously affected and the maximum amplitude during the evolution can be much higher for larger $\sigma_\omega$. In Fig. 4 it is shown that at an influx position ($x \approx -3600$), the maximum amplitudes are quite the same for various fraction of sdv, but at near the focussing position a larger fraction of sdv produces a higher maximum amplitude. This is the consequence of the fact that the larger fraction of sdv gives more wave components with coherent phases.

Although the signal coherence can describe and measure the appearance of freak waves, the concepts use the whole interval of the time signal. However, not the whole interval will contribute in generating a freak wave since the waves propagate with their own group and phase velocity. The freak wave will be generated from local waves interaction. Therefore, we will





investigate the local energy propagation using wavelet transformation. This is expected to give a more refined measure of the appearance of the freak wave.

## 2.2 Wavelet transform

In Fourier analysis we transform a function that depends on time into a function that depends on the frequency as a single variable. Given a time signal $\eta(t)$, Fourier transformation gives the relations,

$$\check{\eta}(\omega) = \frac{1}{2\pi} \int \eta(t) e^{i\omega t} dt$$

$$\eta(t) = \int \check{\eta}(\omega) e^{-i\omega t} d\omega$$

$$= 2 \int_0^\infty |\check{\eta}(\omega)| \cos(\omega t + \theta(\omega)) d\omega.$$

The Fourier transform of $\eta(t)$ is the complex valued function $\check{\eta}(\omega) = |\check{\eta}(\omega)| e^{i\theta(\omega)}$, in which $|\check{\eta}(\omega)|$ is the amplitude spectrum and $\theta(\omega)$ is the phase of the signal. The spectral energy density of the signal is defined by $|\check{\eta}(\omega)|^2$, that describes how the energy of the signal is distributed with frequency. Any local (time) information is not directly contained in Fourier transform, but is hidden in the spectrum and phase. At a certain local time, $t = t_0$, we have

$$\eta(t_0) = 2 \int_0^\infty |\check{\eta}(\omega)| \cos(\omega t_0 + \theta(\omega)) d\omega. \tag{4}$$

The term inside the integral represents the amplitude spectrum and phase distribution with the frequency at a single time. Then we may define a local energy spectrum, $E(t_0, \omega)$,

$$E(t_0, \omega) = (|\check{\eta}(\omega)| \cos(\omega t_0 + \theta(\omega)))^2, \tag{5}$$

presenting the local information of the signal directly. More general, we will not only consider the energy at a single instant, but will also analyze the energy in the neighbourhood. Therefore we will use wavelet transformation for the local energy analysis since it will show the distribution of the local energy spectrum better because it includes energy contributions from neighbouring times instead of only one local time. Figures 5 and 6 illustrate the local energy distribution computed by Fourier and wavelet transform for a dispersive focusing wave and an irregular wave, that will be used as study cases in Sect. 4.1 and 4.3. The plots show that the wavelet transform gives a more refined description of the local energy distribution as function of time and frequency.

The wavelet transform is an extension of Fourier transformation. The basis function in Fourier transform is a sinusoidal of a specific frequency, and the $L^2$-inner product with the signal leads to the Fourier coefficient of that frequency only. A wavelet is composed of a mixture of frequencies (which is indicated by its own Fourier transform). As a consequence, the wavelet coefficients refer to this mixture of frequencies, not a single frequency. We will now provide a summary of the main notions needed in the following sections.



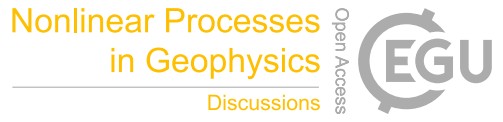

**Definition 2.1.** *A mother wavelet is a zero average function,* $\psi$,

$$\psi \in L^2(\mathbb{R}) : \int \psi(t)dt = 0.$$

**Definition 2.2.** *A wavelet family is family of functions generated from any type of mother wavelet,* $\psi$, *through dilatation* $(s > 0)$ *and translation* $(u \in \mathbb{R})$;

$$\psi_{u,s}(t) = \frac{1}{\sqrt{s}}\psi\left(\frac{t-u}{s}\right).$$

There are many types of mother wavelets: Morlet, Haar, Daubechies, Meyer etc. (see Vialar (2009)). In this paper, we use the Morlet wavelet consisting of a plane wave modulated by a Gaussian, $\psi(t) = e^{-t^2/2}e^{-i\omega_0 t}$ which is given in the Fourier domain by $\hat{\psi}(\omega) = \sqrt{2\pi}e^{-(\omega-\omega_0)^2/2}$ with the central frequency $\omega_0$.

**Definition 2.3.** *The continuous wavelet transform of a signal* $\eta(t)$ *at the scale* $s$ *and at the time* $u$ *is calculated by correlating*
$\eta$ *with the wavelet family,* $\psi_{u,s}$:

$$\mathcal{W}\eta(u,s) = \langle \eta, \psi_{u,s} \rangle = \int \eta(t)\frac{1}{\sqrt{s}}\psi^*\left(\frac{t-u}{s}\right)dt, \tag{6}$$

*where* $\psi^*$ *is the complex conjugate of* $\psi$.

From Definition 2.3, the wavelet transform of a time signal $\eta(t)$ gives a complex valued function $\mathcal{W}\eta(u,s)$. For the Morlet wavelet, we obtain

$$\mathcal{W}\eta(u,s) = \int \eta(t)\frac{1}{\sqrt{s}}e^{-(t-u)^2/2s^2}e^{i\omega_0(t-u)/s}dt.$$

By substituting $s = \omega_0/\omega$ and writing $F(t; u, \omega) = \frac{1}{\sqrt{s(\omega)}}e^{-(t-u)^2/2s(\omega)^2}$, the equation above gives

$$\mathcal{W}\eta(u,\omega) = \int \eta(t)F(t; u, \omega)e^{i\omega(t-u)}dt.$$

The function $F(t; u, \omega)$ applies as a Gaussian window function to the signal $\eta(t)$. This shows that the wavelet transform can be interpreted as the Fourier transform of a windowed signal in the neighbourhood of $t = u$. The magnitude of the wavelet
transform, $|\mathcal{W}\eta(u,\omega)|$, represents the energy distribution of the signal over frequency and time and its angle, $\arg(\mathcal{W}\eta(u,\omega))$, represents the local phase of the signal.

Similar to Fourier transform, it is possible to rebuild the signal from the wavelet transform, the so-called inverse wavelet transform. It is given by:

$$
\begin{aligned}
\eta(t) &= \frac{1}{C_\psi} \int_0^\infty \int \mathcal{W}\eta(u,s)\frac{1}{\sqrt{s}}\psi\left(\frac{t-u}{s}\right)du\frac{ds}{s^2} \\
&= \int_0^\infty \left[\frac{1}{\omega_0 C_\psi} \int \mathcal{W}\eta(u,\omega)F(t; u, \omega)e^{-i\omega(t-u)}du\right]d\omega
\end{aligned}
\tag{7}
$$



with

$$C_\psi = \int\limits_0^\infty \frac{|\hat{\psi}(\omega)|^2}{\omega} d\omega.$$

As an example, for $\omega_0 = 6$ the Morlet wavelet above produces $C_\psi \approx 1.883$. Different from equation (4) that gives the local energy spectrum computed at one time, equation (7) shows that the local energy spectrum from the wavelet transform is not

only computed at the local time, but it also includes the contribution of the signal surrounding that time.

The choice of the central frequency $\omega_0$ should be such that the Morlet wavelet satisfies the *admissibility* condition, $C_\psi < \infty$, which is equivalent to $\hat{\psi}(0) = 0$. Then the (real) wavelet transform is complete and preserves the quantity of energy;

$$\int |\eta(t)|^2 dt \approx \frac{1}{\omega_0 C_\psi} \int\limits_0^\infty \int |\mathcal{W}\eta(u,\omega)|^2 du d\omega.$$

Actually, the Morlet wavelet satisfies the condition only approximately because $\hat{\psi}(0) = \sqrt{2\pi} e^{-\omega_0^2/2}$ does not vanish exactly.

A proper choice of $\omega_0$ can make the wavelet at least practically admissible and allows one to apply it widely to the signal decomposition (Lebedeva and Postnikov, 2014; Mertins, 1999). The defined Morlet wavelet is sufficiently admissible if we choose $\omega_0 > 5$ (see Mertins (1999)), hence $\omega_0 = 6$ is taken to be sufficient since then $\hat{\psi}(\omega) \le 3.8 - E8$, for $\omega \le 0$. Parseval's identity gives a relation between the signal and its Fourier transform as

$$\begin{aligned} \int |\eta(t)|^2 dt =& 2\pi \int |\breve{\eta}(\omega)|^2 d\omega \\ \approx& \int \left[ \frac{1}{\omega_0 C_\psi} \int |\mathcal{W}\eta(u,\omega)|^2 du \right] d\omega. \end{aligned}$$

Therefore, the spectral energy density of a signal can be computed through the wavelet transform, i.e

$$|\breve{\eta}(\omega)|^2 \approx \frac{1}{2\pi \omega_0 C_\psi} \int |\mathcal{W}\eta(u,\omega)|^2 du. \tag{8}$$

This equation shows that the energy distribution from the wavelet transform behaves locally, and its integration over the time shift $u$ is approximately the spectral energy density obtained by Fourier transform.

## 3   Characterizing Freak Waves

The capability of the wavelet transform to represent a signal in time and frequency domain motivates us to investigate a freak wave locally. For a given signal, we identify group events which are parts of the time signal that may develop into propagating wave groups, i.e. that contain an amount of energy larger than a certain threshold. This threshold is determined such that the group event can build a freak wave if additional conditions are satisfied. We then determine the most energetic waves from each group event to see how the energy is distributed in both time and frequency. The most energetic waves will determine the

evolution of the group event, and whether its energy will converge or diverge. With these elements, we will be able to define the local coherence which will describe quantitatively the process of freak wave formation from a critical group event.





### 3.1 Critical group events

Holthuijsen (2007) defines a wave group as an uninterrupted sequence of waves with wave heights higher than an arbitrarily chosen, but usually high, threshold value. Instead of a wave group, we define a group event based on a chosen local energy level as threshold, which is determined by the contour level of the spectral energy determined by the wavelet transform. A group event of a time signal $\eta(t)$ is part of the time signal with $|\mathcal{W}\eta(u,\omega)|$ higher than a threshold value. We denote the set of group events with respect to the threshold value $\epsilon$, by $WG_\epsilon$,

$$
\begin{aligned}
WG_\epsilon &= \{\eta_i(t) : i = 1, 2, \cdots, N_g\} \\
\eta_i(t) &= \{\eta(t) : t = [t_1, t_2] \subset [0, T] \,|\, |\mathcal{W}\eta_i(u,\omega)| \geq \epsilon\}
\end{aligned}
\tag{9}
$$

$WG_\epsilon$ is the assembly of $N_g$ group events; each group is determined by the time interval during which the wavelet transform is larger than a specified value $\epsilon$. The selection of the group events depends on the chosen threshold value $\epsilon$. In practice, we normalize the value of $|\mathcal{W}\eta|$ with its maximum, so that the value of $\epsilon$ is chosen in $(0, 1)$. The choice depends on the background waves since it aims to ignore the waves that do not contribute to the evolution of the group under consideration. When the background waves are high, we should choose a large $\epsilon$, but when the background waves are small, we can choose a small value of $\epsilon$. In this paper, we choose $\epsilon \approx 0.65$ for the random signals and $\epsilon \approx 0.2$ for the maximal signal.

From all the group events determined in this way, we characterize the groups that may lead to a freak wave. For a given time signal, $\eta(t)$, $t \in [0, T]$, we define a total energy signal, $E_T$,

$$
E_T = \int_0^T |\eta(t)|^2 dt \approx \frac{1}{\omega_0 C_\psi} \int_0^T \int |\mathcal{W}\eta(u,\omega)|^2 d\omega du.
\tag{10}
$$

For each value of the total energy signal, there can be a maximal wave with a coherent state.

Next we define the total energy threshold to eliminate group events which unlikely generate a freak wave. The remaining groups are so-called *critical group events*.

$$
WG_{crit} = WG_\epsilon \cap \left\{ \eta_i(t) \,\middle|\, \int_{t_1}^{t_2} \eta_i(t)^2 \, dt \geq \rho^2 E_T \right\}
\tag{11}
$$

in which

$$
\rho = \frac{1.25 H_s}{\displaystyle\int |\check{\eta}(\omega)| d\omega}
$$

is a freak wave threshold normalized by the amplitude of a maximal signal. Based on their local energy, these critical group events could generate a freak wave forward or backward, but the probability depends on the phases.





### 3.2 Most energetic waves

We start from the complex value of the wavelet transform of $\eta(t)$,

$$\mathcal{W}\eta(u,\omega) = |\mathcal{W}\eta(u,\omega)|e^{i\Theta(u,\omega)}.$$

It gives the spectral energy distribution $|\mathcal{W}\eta(u,\omega)|$ and the phase information $\Theta(u,\omega)$ as a function of time and frequency. From this we may look at the frequencies that carry most energy as a function of time denoted by $\omega_m(u)$ from the critical group event,

$$\omega_m(u) = \left\{\omega \,\middle|\, |\mathcal{W}\eta(u,\omega)| = \max_\omega |\mathcal{W}\eta(u,\omega)| \right\}$$

Convergence of waves will occurr when long waves catch up with shorter waves. Hence, when the local wave length increases, i.e. when the wave frequency decreases, the waves will converge at a later time and vice versa. Therefore, the distinction is determined by the frequency in the time interval: when decreasing in forward time, this leads to a focussing energy, and an increase leads to de-focussing energy. Since continuity of the local wave frequency in the random waves cannot be guaranteed, we approximate the local wave frequency by a linear interpolation, $\omega_m(u) \approx \tilde{\omega}(u)$, so that we can distinguish the two cases

$$\tilde{\omega}(u) = \tilde{A}u + \tilde{B} \begin{cases} \tilde{A} > 0, & \text{de-focussing/ diverging energy} \\ \tilde{A} \leq 0, & \text{focussing/ converging energy} \end{cases} \tag{12}$$

Moreover we can also look at the most energetic waves as a function of wave frequency. This leads to a local time of each wave contribution. In the case of a dispersive focussing wave, focussing of the energy occurs when all wave contributions are in phase at one local time.

Motivated by this, for each critical group event in a local time interval $[t_1, t_2]$, we define a function $\tau_m(\omega)$ representing the local time of the maximal energy, $\tau_m(\omega) \in [t_1, t_2]$,

$$\tau_m(\omega) = \left\{ u \,\middle|\, |\mathcal{W}\eta(u,\omega)| = \max_u |\mathcal{W}\eta(u,\omega)| \right\} \tag{13}$$

Hence, if the critical group event gives a constant $\tau_m(\omega)$, all frequencies contribute at the same time which leads to local coherence at that time. If the frequencies are decreasing over the local time interval, it may indicate a local focussing at a later time.

### 3.3 Local coherence

The observations of the most energetic waves in either time or frequency can be used to see whether a freak wave may appear in forward or backward time, but the generation of a freak wave is still not assured, since the amplitude is not determined yet. The local information of the energy and phase gives a method to investigate locally the relation between the local coherence and freak wave occurrence. In this subsection, we measure the local coherence of the group event along its evolution and we will show that the highest amplitude occurs when the local coherence is maximum in the restricted frequency interval. As





the wavelet transformation gives a function of frequency and time, we define a time spreading of the most energetic waves $[\varphi(\omega)]_\tau \in [-\pi, \pi]$ for each time $\tau \in (t_1, t_2)$ as follows:

$$[\varphi(\omega)]_\tau = [\tau_m(\omega) - \tau] \bmod 2\pi$$

that is taken at the time at which the absolute mean is minimal.

$$\varphi(\omega) = \left\{ [\varphi(\omega)]_\tau \,\middle|\, |\overline{\varphi}(\omega)|_\tau = \min_\tau |\overline{\varphi}(\omega)|_\tau \right\} \tag{14}$$

The time spreading is exactly zero at a certain frequency interval when $\tau_m(\omega)$ is constant at that interval. To investigate the local coherence, we determine the maximum $(M)$, the mean $(\mu)$, and the standard deviation $(\sigma)$ of the absolute value of the time spreading normalized by $\pi$. Accordingly, we define three quantities depending on position that can represent local coherence, $\Gamma_{M,\mu,\sigma}(x) \in [0,1]$, depending on the choice for the parameters, $M, \mu$ or $\sigma$:

$$\Gamma_M(x) = 1 - M \qquad\qquad \Gamma_\mu(x) = 1 - 2\mu \qquad\qquad \Gamma_\sigma(x) = 1 - \sqrt{3}\sigma \tag{15}$$

These values represent a somewhat different measure of local coherence. Note that the extreme case $(\Gamma_M = \Gamma_\mu = \Gamma_\sigma = 1)$ occurs for the maximal signal, when all the phases are zero. Note also that this measure is different from the degree of phase coherence defined in Latifah and van Groesen (2012) as it corresponds to the *local time spreading* of the most energetic waves of a group event. To investigate the dependence between the local coherence and the occurrence of freak waves, we com-
pute the correlation between the local coherence and the maximum amplitude normalized by its time averaged local energy, $Corr(\Gamma_{M,\mu,\sigma}, A_m)$. For $N$ number of time signals at the positions $(x_1, x_2, \cdots, x_N)$, the correlation is computed by

$$Corr(\Gamma, A_m) = \frac{\sum\limits_{i=1}^{N} (\Gamma(x_i) - \mu_\Gamma)(A_m(x_i) - \mu_A)}{(N-1)\sigma_\Gamma \sigma_A} \qquad \text{with} \qquad A_m(x_i) = \frac{\max\limits_{t \in (t_1, t_2)} |\eta(x_i, t)|}{\dfrac{1}{t_2 - t_1} \int\limits_{t_1}^{t_2} \eta(x_i, t)^2 dt}$$

and $\Gamma(x_i)$ is the local coherence of the time signal at $x_i$.

## 4 Case Studies

This section presents the investigations of four study cases: an experimental dispersive focussing wave, a synthetic normal wave condition (W100), a synthetic thunder storm condition (TS10000), and an experimental irregular wave (IW12). For each case, we start to characterize the critical group events, then we investigate the local features of these groups, namely the most energetic wave and its time spreading. We investigate the evolution of the local energy and the time spreading of each case, particulary around the critical group events, and measure the local coherence. Furthermore, we compute the correlation between
the local coherence and the maximum amplitude of the group event that generates a freak wave. It will give an impression of the relevance of the parameters $\Gamma$ for measuring a freak wave.

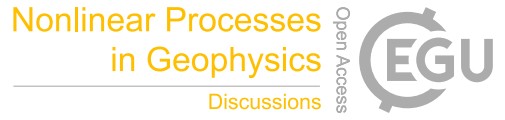

### 4.1 Focussing wave (202002)

The case is a focussing wave that will lead to a maximal wave. We consider a dispersive focussing wave, for which measurements at several positions are available from an experiment at MARIN (Case 202002). The experiment was executed at a water depth of 1m. Here we use the elevation at the first measurement position after the wave flap as the influx signal for the numeri-

cal simulation by both the linear and nonlinear AB-equation. The result of the numerical simulation of a dispersive focussing wave using both the linear and nonlinear AB-equation have been previously verified with the measurements (Liam et al., 2014; Lakhturov et al., 2012).

Referring to Fig. 5 the influx signal only consists of one group event with almost zero background which is therefore the only critical group event. This is an idealized case as the freak wave turns out to be a maximal wave that is generated from all

wave components in the initial signal. This can be observed from the evolution of the influx signal, the shorter (slower) waves are followed by longer (faster) waves such that at the focussing point at 50.2m all waves have vanishing phase. See Fig. 7 for various plots of snapshots of the dynamics at successive measurement positions.

During the evolution, the changes of the distribution of the local energy in time-frequency frame are described well by the filled countour plot of the local energy. The local energy distribution from one group event is squeezed into a maximal wave.

This is also shown by the decreasing width of the time intervals towards the focussing point in Fig. 8. We can see at $x = 20$m that the energy is distributed in 20s, at $x = 40$m it is distributed approximately in 10s and at the focussing point the energy is only distributed in 3s. Moreover, the pure maximal wave is shown by the zeroes of the time spreading at $x = 50.2$m in Fig. 7b. The profile of the maximal wave can be seen in Fig. 9.

In order to show that the occurrence of the freak wave is related to a local coherence, and to illustrate the three different

measures of coherence introduced above, we show the evolution of these coherence measures for the linear and nonlinear evolution in Table 1. It can be observed from this table that the correlation of each of the three measures of coherence and the occurrence of the maximum amplitude at $x = 50.2$m is very strong ($\approx 0.95$), although outside the focussing position the values of the three $\Gamma$'s can be rather different. $\Gamma_M$ and $\Gamma_\mu$ seem to be much better indicators for the focussing than $\Gamma_\sigma$.

### 4.2 Synthetic signals

The second and third case are synthetic signals of irregular waves, that are generated from a Jonswap spectrum with normal and thunder storm sea conditions at a water depth of 480m (deep water). The wave evolutions are computed linearly by the AB-equation as the nonlinear effect for these cases is not significant. However, a freak wave is still found in both cases.

#### 4.2.1 Normal sea (W100)

The initial time signal is generated from a Jonswap spectrum with time period 11.3s, $\gamma = 1.9$, and significant wave height 6.3m

(van 't Veer and Vlasveld, 2014). The duration of the time signal is approximately 3 hours. From the initial time signal, there are nine critical group events, of which the two largest groups will be investigated. We do not investigate the other critical group events since their amount of the local energy signal are slightly equal to the threshold such that they are unlikely to develop a





freak wave. Figure 10 shows the two critical groups of the influx signal with approximately the same amount of local energy signal; one is around $t = 3200$s and the other is at $t \approx 3600$s. Those are the most probable group events that can develop a freak wave. In the observation of the contour energy distribution, the preceding group event gives positive $\tilde{A}$ while the other one gives negative value. Therefore, the critical group event around $t = 3600$s is the candidate to generate a larger amplitude

in forward time. The evolution of this critical group together with its energy distribution is shown in Fig. 11a and the changes of its time spreading is in Fig. 11b. We observe that at the freak wave position ($x = 1420$m), the time spreading is almost zero for the wave carrying modes. Outside the freak wave position, the time spreading of the critical group event is distributed in $[0, \pi]$. The freak wave is shown in Fig. 12.

In this case, the occurrence of the freak wave can also be observed from the most energetic wave in either time or frequency

(see Fig. 13). Before the freak wave, the most energetic waves give a decreasing wave frequency and after the freak wave, an increasing wave frequency occurs. At $x = 1420$m, the local time of the maximal energy is almost constant for the carrying wave modes ($\omega \in [0.5; 0.7]$), therefore its time spreading is nearly coherent and it generates a freak wave.

Furthermore, we investigate the change of the local coherence of the critical group event during its 3km linear wave evolution. The measure of coherence at various positions is shown in Table 2 and the correlation between the local coherence and

the maximum amplitude along the evolution is presented in the lowest row. All three $\Gamma$'s show a quite high correlation ($\geq 0.74$) between the local coherence and the maximum amplitude. According to the correlation value, $\Gamma_M$ and $\Gamma_\mu$ seem to be better indicators for the freak wave appearance than $\Gamma_\sigma$.

### 4.2.2 Thunder storm sea (TS10000)

The other synthetic signal is generated from a Jonswap spectrum with time period 13.6s, $\gamma = 2$, and significant wave height

15.2m (van 't Veer and Vlasveld, 2014). A snapshot of the initial time signal is shown in Fig. 14. This type of wave is categorized as thunder storm sea condition, in which the appearance of a freak wave is more probable than in a normal sea condition. The duration of the initial time signal is approximately 3 hours. There are five critical group events found from the influx signal, but the two unlikely ones do not generate a freak wave since their local energy signal is not so high compared to the threshold. The largest local energy signal of the group events appears around $t \approx 5400$s and its maximum crest is already quite high at

the initial time. Then in forward time it still develops to a higher crest and generates a freak wave.

Figure 15a presents the snapshots of the time signals at various positions. Also shown is the local energy distribution of the critical group event that leads to a freak wave. In Fig. 15b, the time spreading of the critical group event shows the chosen carrying wave modes ($\omega \in [0.45; 0.52]$). A freak wave appears at $x = 2985$m (see Fig. 16). If we observe the time spreading at $x = 2000$m, it seems that the local time is more coherent than at the freak wave position. This can also be seen from the

measure of the local coherence in Table 3. The larger amplitude of the freak wave compared to the group event at $x = 2000$m can be explained from its local energy distribution. The width in time of the energy spectral distribution is a bit squeezed and there is some higher wave frequency contribution which does not appear at $x = 2000$m. Figure 17 shows the filled contour plot of the local energy distribution for the most energetic waves at several positions as function of time and frequency. It can be



observed that there is a change of the wave frequency order. Before the freak wave, the short waves run ahead the long waves and after the freak wave, the short waves are behind, just as in focussing waves.

We measure the local coherences of the critical group event along its linear evolution and the results are presented in Table 3. The correlation for each local coherence $\Gamma$ and the maximum amplitude normalized by the local energy signal is quite high ($\geq 0.78$). This shows that the appearance of the freak wave is mostly caused by the local coherence of the critical group event from the influx signal.

### 4.3 Experimental signal: Irregular Wave (IW12)

The fourth case is an irregular wave, for which measurements at several positions are available from MARIN experiment (Case 103001). Compared to the laboratory experiment the spatial dimension was scaled by the water depth of 50m (1:50 in space) to geophysical dimensions. It has 12s peak period. The duration of each measurement is approximately 3 hours. We only use the first 1 hour time signal from the first measurement position after the wave flap as the influx signal. The local energy distribution of the signal is presented in Fig. 6. There are six critical group events from the influx signal as shown in Fig. 18. The largest local energy signal of the wave groups is found around $t \approx 1700s$ and it develops a freak wave.

The evolution of the time signal around the critical group event and its energy distribution at several positions are shown in Fig. 19a. Eventhough the energy spectral distribution does not show clearly the development of the critical group event into a freak wave, the change of the time spreading shows the development of its local coherence (see Fig. 19b). A freak wave occurs at $x = 5185m$ when its time spreading is near coherent for a short carrying wave mode. The freak wave is shown in Fig. 20. From Fig. 21 we can also see that there is unclear increasing or decreasing wave frequencies of the most energetic wave. The local coherences are measured and presented in Table 4. The three values of $\Gamma$'s present quite high correlation ($\geq 0.75$) between the local coherences and the maximum amplitude in both the linear and nonlinear evolution. In this case $\Gamma_\mu$ performs as the best indicator for the freak wave appearance.

### 5 Conclusions

In this paper we showed the relevance of phase coherence by illustrations of signals with increasingly less restrictions on the phase function. Then the wavelet transform is used to determined the time-frequency spectrum of a time signal. We used the wavelet transform to identify critical group events of the influx signal and it is shown that the group event with the largest local energy signal is the most probable group to generate a freak wave. We remarked that the identification of a group event is depending on the choice of the threshold value ($\epsilon$). For irregular waves, we suggested to choose $\epsilon \approx 0.65$ and for waves with vanishing background we could choose a smaller value $\epsilon \approx 0.2$. We defined local coherence by three parameters (the mean, maximum or standard deviation) of the time spreading of the most energetic waves from the critical group events. We investigated the change of the local coherence along its evolution and showed that all three values of the local coherence are strong indicators for the appearance of a freak wave. This indicates a local mechanism of a freak wave appearance: the freak wave is mostly developed by a local coherence of a group event. At the influx signal, the group event already contains





a considerable amount of energy, which evolves into successive states with even higher coherence. Four study cases illustrate the usefulness of the introduced concepts to describe and predict the appearance of freak waves.

*Acknowledgements.* This work was funded by the Netherlands Organization for Scientific Research, Technology foundation STW, number 7216. We acknowledge MARIN hydrodynamic laboratory for their measurement data 202002 and 103001 used in this paper.

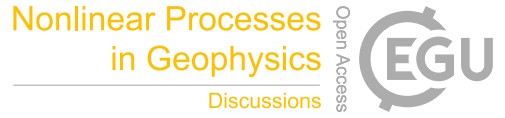

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





**Table 1.** Measure of the local coherence of the dispersive focussing wave

| $x$ | Linear | | | | | Nonlinear | | | | |
|---|---|---|---|---|---|---|---|---|---|---|
| | $\Gamma_M$ | $\Gamma_\mu$ | $\Gamma_\sigma$ | $A_m$ | $\tilde{A}$ | $\Gamma_M$ | $\Gamma_\mu$ | $\Gamma_\sigma$ | $A_m$ | $\tilde{A}$ |
| 20 | 0.009 | 0.044 | 0.504 | 0.012 | - | 0.002 | 0.009 | 0.04 | 0.49 | - |
| 30 | 0.002 | 0.116 | 0.516 | 0.019 | - | 0.001 | 0.002 | 0.12 | 0.52 | - |
| 40 | 0.001 | 0.231 | 0.507 | 0.046 | - | 0.001 | 0.001 | 0.23 | 0.51 | - |
| 45 | 0.312 | 0.285 | 0.659 | 0.113 | - | 0.293 | 0.29 | 0.27 | 0.65 | - |
| 50.05/ 50.2 | 0.987 | 0.996 | 0.994 | 0.692 | + | 0.975 | 0.98 | 0.97 | 0.99 | + |
| 56 | 0.230 | 0.208 | 0.627 | 0.097 | + | 0.281 | 0.28 | 0.24 | 0.64 | + |
| Corr($A_m(x),\Gamma$) | 0.95 | 0.96 | 0.94 | 1 | | 0.93 | 0.94 | 0.92 | 1 | |





**Table 2.** Measure of the local coherence of the normal sea condition wave

| $x$ | $\Gamma_M$ | $\Gamma_\mu$ | $\Gamma_\sigma$ | $A_m$ | $\tilde{A}$ |
|---|---|---|---|---|---|
| 500 | 0.05 | 0.10 | 0.35 | 0.01 | - |
| 1420 | 0.68 | 0.90 | 0.76 | 0.018 | - |
| 2000 | 0.045 | 0.167 | 0.397 | 0.010 | - |
| 2500 | 0.045 | 0.175 | 0.306 | 0.008 | + |
| 3000 | 0.045 | 0.028 | 0.341 | 0.007 | + |
| $\mathrm{Corr}(A_m(x),\Gamma)$ | 0.78 | 0.82 | 0.74 | 1 | |





**Table 3.** Measure of the local coherence of the thunder storm condition wave

| $x$ | $\Gamma_M$ | $\Gamma_\mu$ | $\Gamma_\sigma$ | $A_m$ | $\tilde{A}$ |
|---|---|---|---|---|---|
| 1500 | 0.19 | 0.17 | 0.52 | 0.01 | - |
| 2000 | 0.53 | 0.66 | 0.77 | 0.018 | - |
| 2985 | 0.37 | 0.39 | 0.69 | 0.010 | - |
| 3500 | 0.05 | 0.03 | 0.47 | 0.008 | + |
| 3800 | 0.11 | 0.16 | 0.50 | 0.007 | + |
| Corr$(A_m(x),\Gamma)$ | 0.80 | 0.78 | 0.81 | 1 | |





**Table 4.** Measure of the local coherence of IW12

| $x$ | Linear | | | | | Nonlinear | | | | |
|---|---|---|---|---|---|---|---|---|---|---|
| | $\Gamma_M$ | $\Gamma_\mu$ | $\Gamma_\sigma$ | $A_m$ | $\tilde{A}$ | $\Gamma_M$ | $\Gamma_\mu$ | $\Gamma_\sigma$ | $A_m$ | $\tilde{A}$ |
| 4000 | 0.19 | 0.31 | 0.46 | 0.005 | - | 0.325 | 0.511 | 0.618 | 0.005 | - |
| 4500 | 0.15 | 0.46 | 0.49 | 0.007 | - | 0.415 | 0.588 | 0.666 | 0.006 | - |
| 5110/5185 | 0.82 | 0.86 | 0.90 | 0.009 | - | 0.685 | 0.820 | 0.814 | 0.009 | - |
| 5500 | 0.55 | 0.66 | 0.74 | 0.006 | + | 0.775 | 0.833 | 0.869 | 0.006 | + |
| 6000 | 0.10 | 0.22 | 0.41 | 0.006 | + | 0.415 | 0.331 | 0.637 | 0.006 | + |
| $\mathrm{Corr}(A_m, \Gamma)$ | 0.78 | 0.86 | 0.78 | 1 | | 0.75 | 0.88 | 0.76 | 1 | |





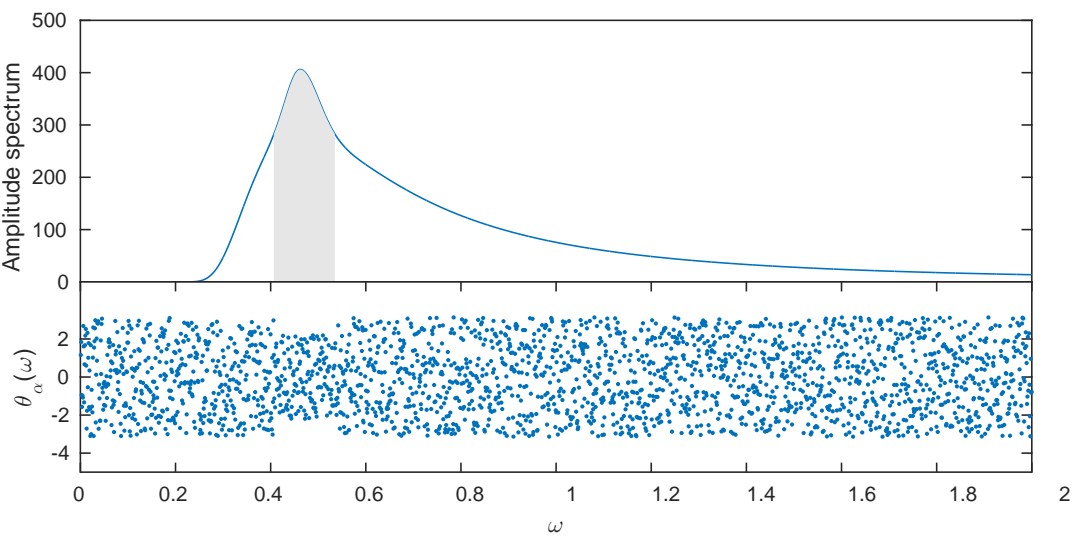

**Figure 1.** A Jonswap spectrum with restricted random phases, $\theta_\alpha(\omega)$. The shaded area represents the energy carrying modes (restricted by a half standard deviation).



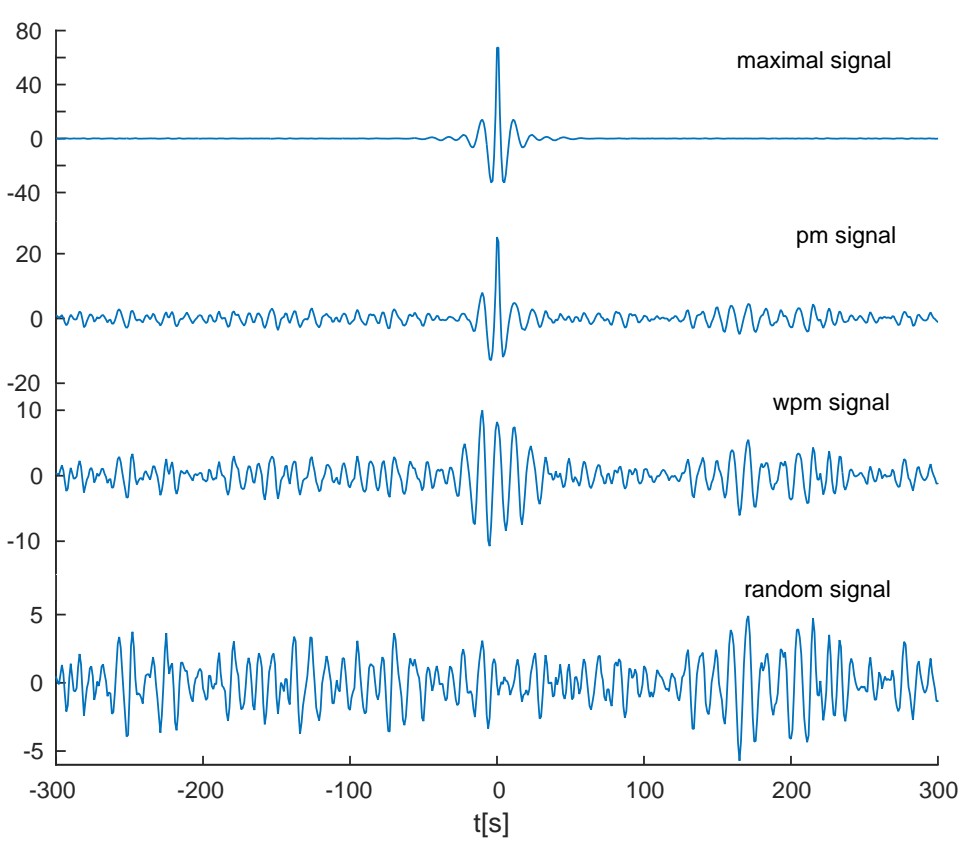

**Figure 2.** Shown are plots from up to down of a maximal, pseudo-maximal, and weak-pseudo maximal signal corresponding to the same random signal at the bottom. The random signal corresponds to a Jonswap spectrum with $H_s = 6.3$m. The pm and wpm signals correspond to the value $\alpha = 0.7$.

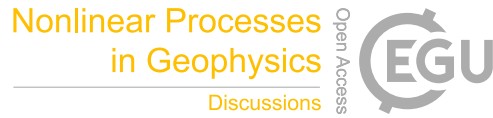



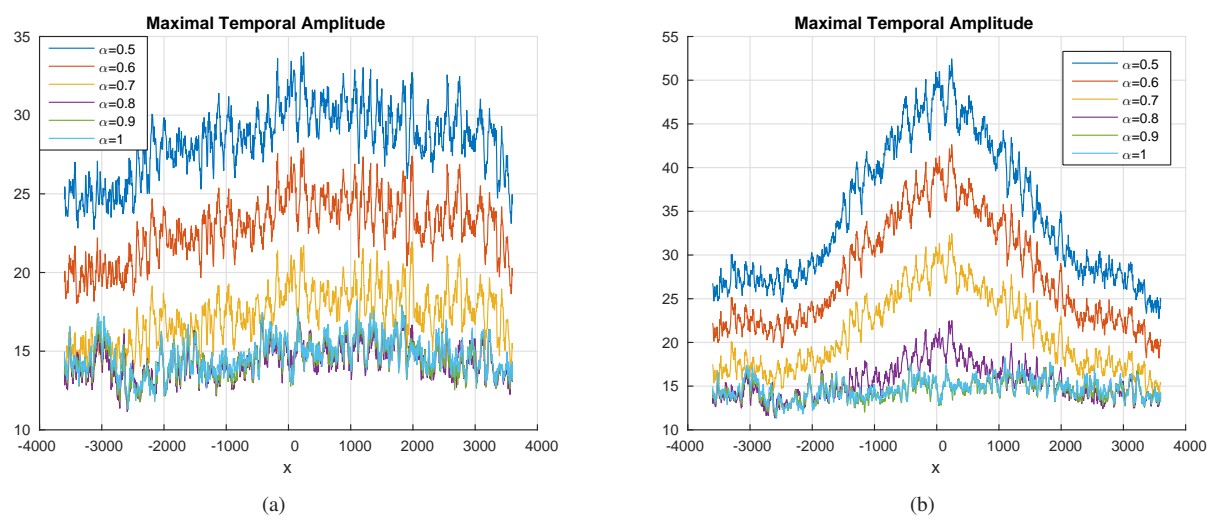

(a)                                                 (b)

**Figure 3.** The maximal temporal amplitude of the linear evolution of the wpm signal for various values of $\alpha$. Figure (a) corresponds to restricting the phases to a quart sdv, $|\omega - \omega_p| < 0.25\sigma_\omega$ and (b) for a half sdv, $|\omega - \omega_p| < 0.5\sigma_\omega$.



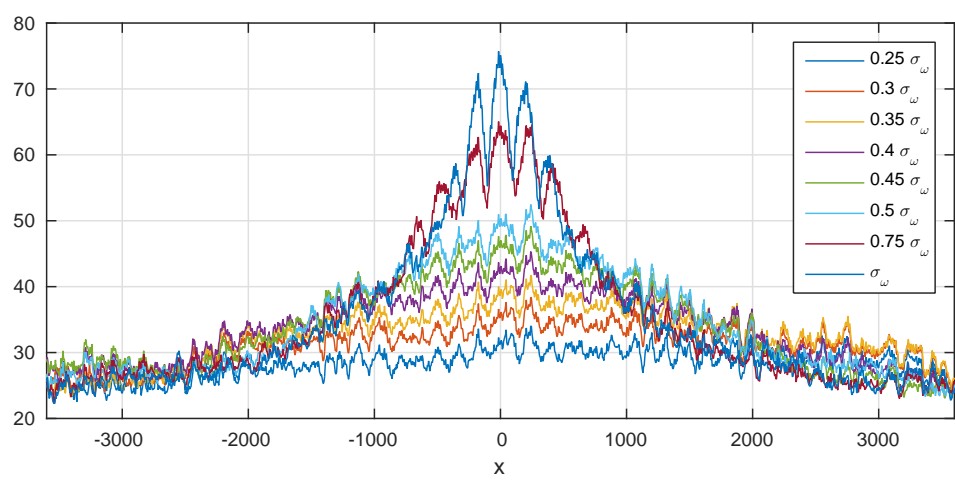

**Figure 4.** The maximal temporal amplitude of the linear evolution of the wpm signal with $\alpha = 0.5$ for various fractions of the standard deviation $\sigma_\omega$.

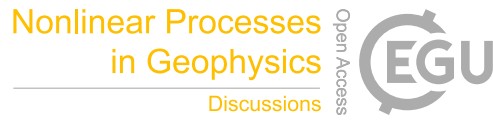



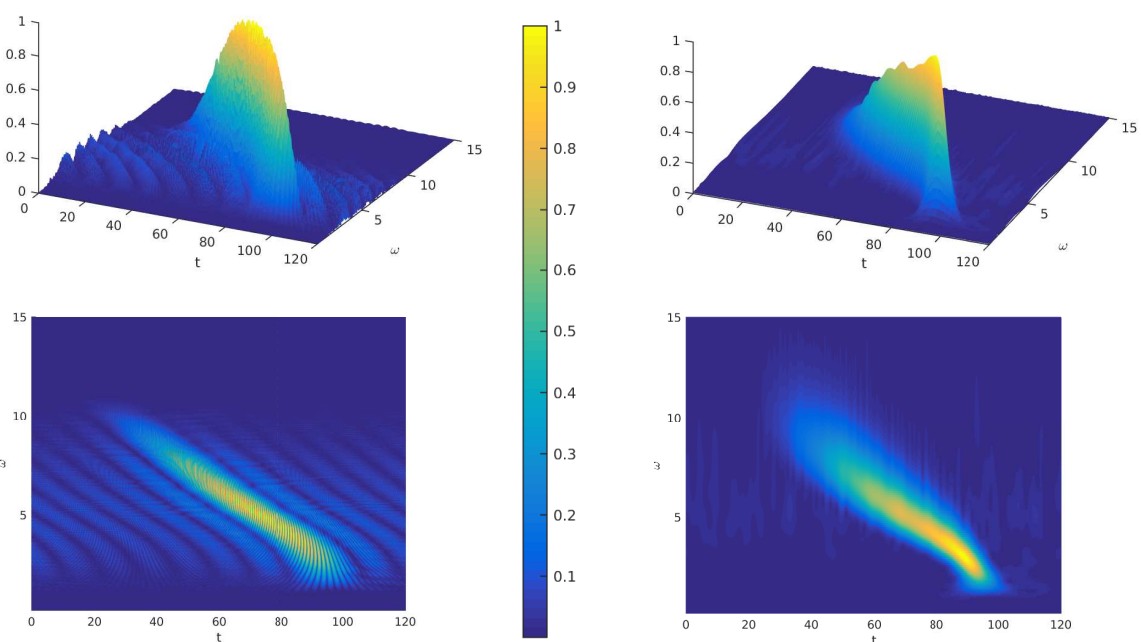

**Figure 5.** Shown is the distribution of the local energy of the dispersive focusing wave at some positions before the focusing point. The left plots are computed by Fourier transform and the right plots are by wavelet transform. The upper plots are in 3D-view, while the lower plots are in 2D-view.





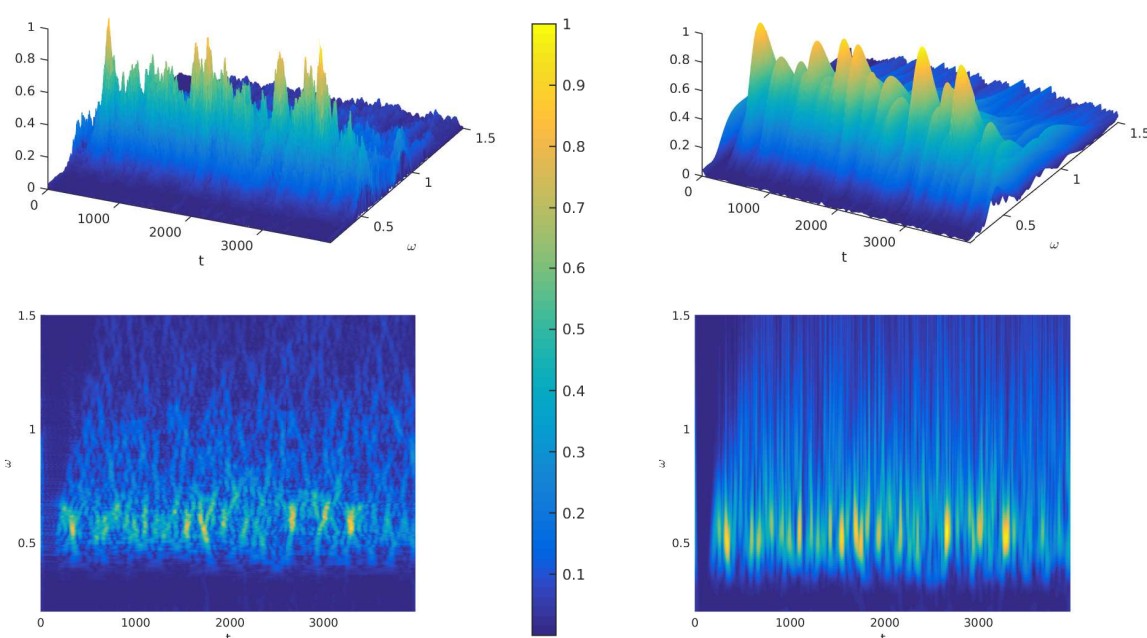

**Figure 6.** The same as Figure 5, now for the irregular wave IW12 at some positions before the freak wave.





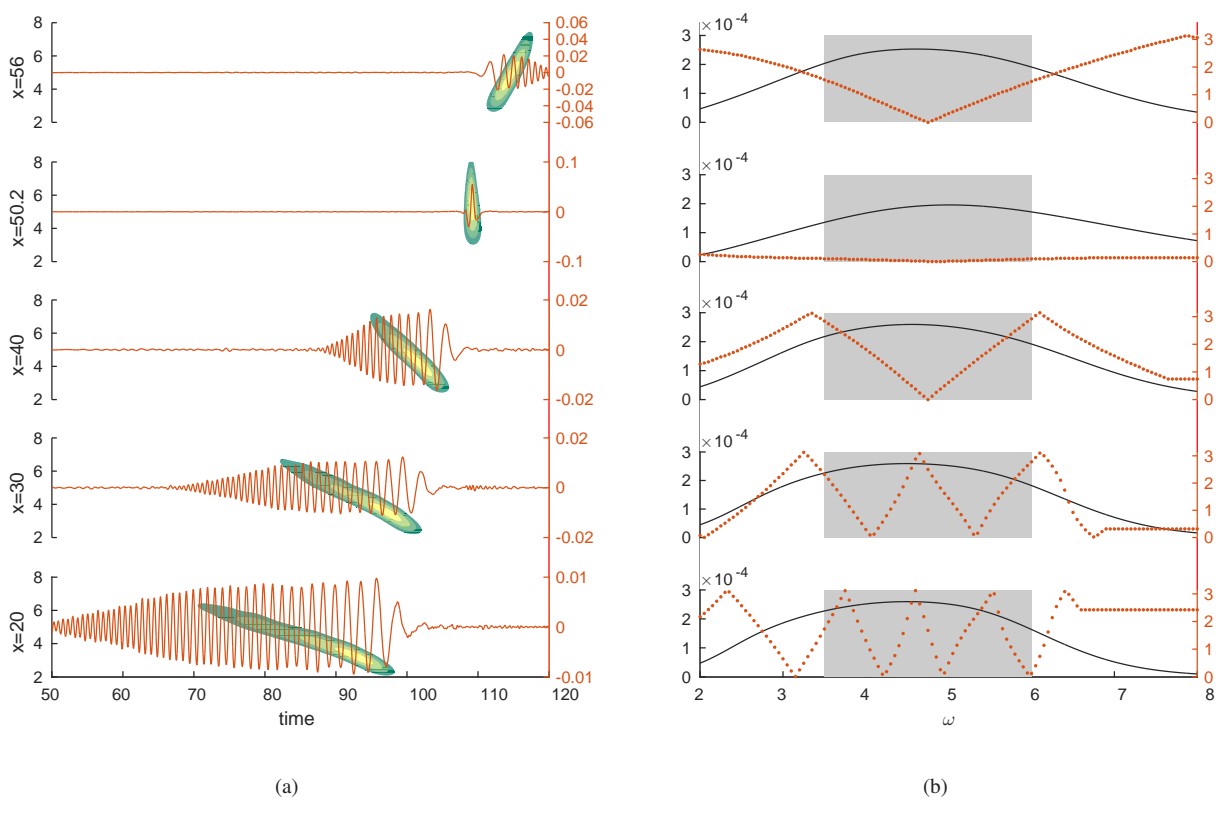

|     |     |
| :-: | :-: |
| (a) | (b) |

**Figure 7.** Case 202002. (a) Time signals at various positions of the evolution of the critical group event with the filled contour plot of wavelet spectra. The vertical axis at the left represents the wave frequency $\omega$ and the vertical axis at the right represents the surface elevation in meters. (b) The corresponding time-averaged wavelet spectra (solid line) and the time spreading (dotted line). Observe that at $x = 50.2$ the time spreading vanishes identically in the shaded area. The shaded areas show the chosen frequency interval of the most energy carrying modes.





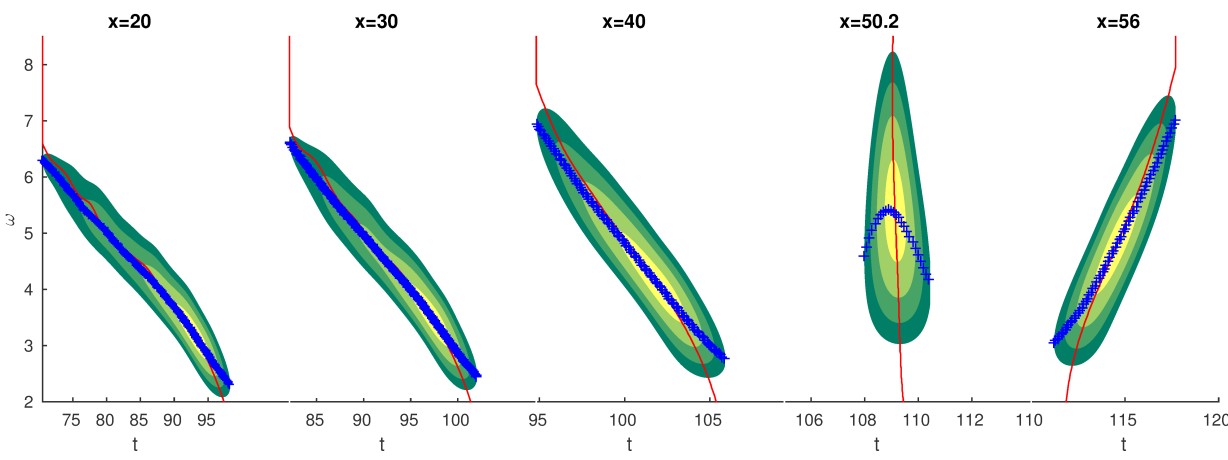

**Figure 8.** Case 202002. A filled contour plot of the energy distribution of the critical group event at position $x = 20, 30, 40, 50.2, 56$. At each position the red solid lines show the time of maximal energy at each wave frequency. The '++' lines show the wave frequency as function of time. Both are estimated by the most energetic waves in time and in frequency respectively. Before $x = 50.2$ both solid and '++' lines show decreasing frequencies (increasing wave length) in time, then it leads to energy convergence.





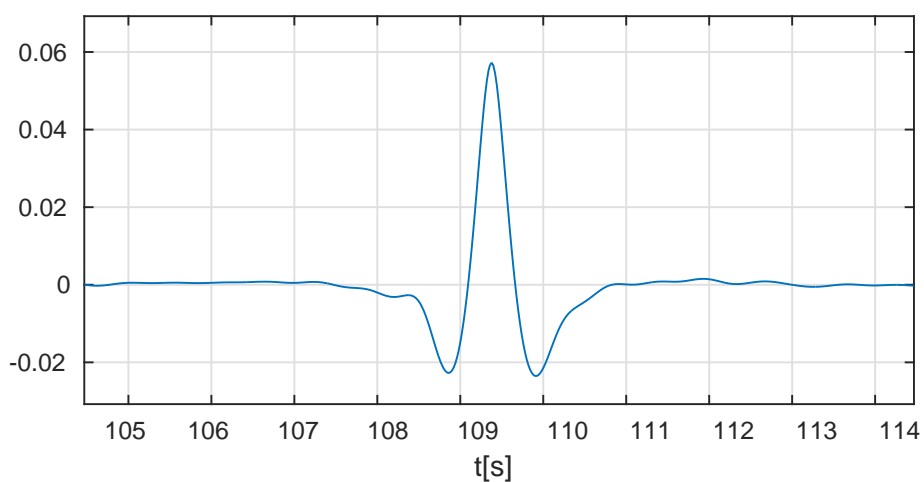

**Figure 9.** Case 202002. Zoomed version of the maximal wave; the crest height is 4.65 and the wave height is 6.56 times the significant wave height.





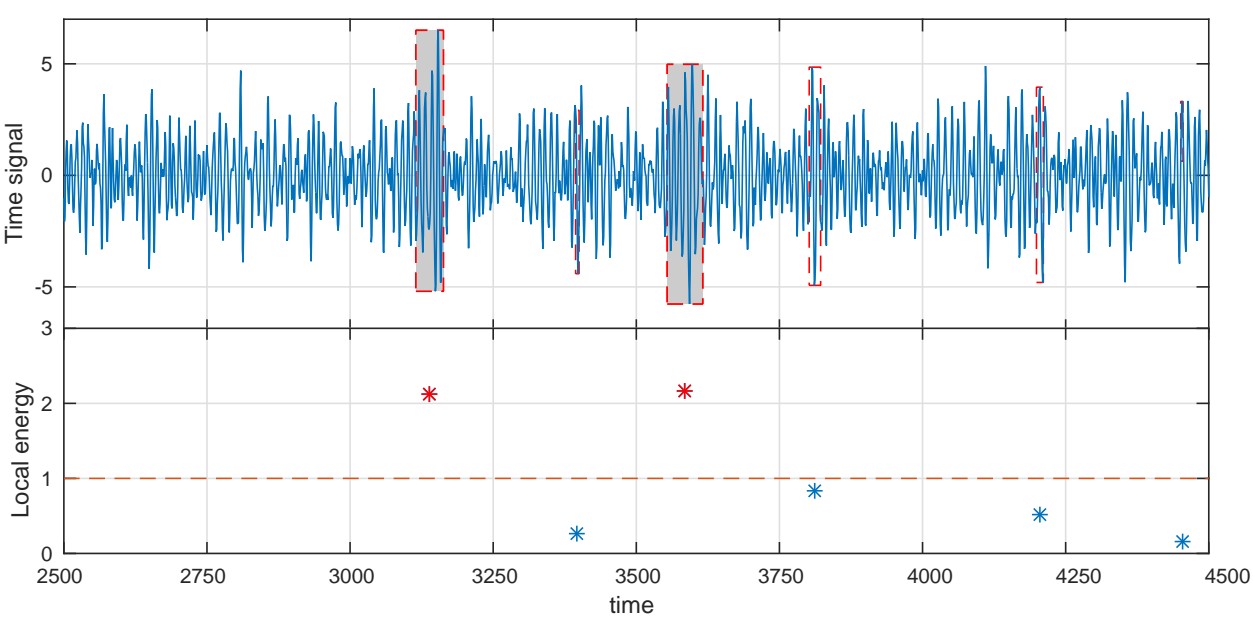

**Figure 10.** Case W100. Initial time signal in the interval $t \in [2500, 4500]$. The critical group events are shown in the shaded areas of the upper plot. The lower plot presents the amount of local energy signal of the recognized group events compared to the local energy threshold (dashed line). The local energy signal of the critical group events are above the threshold.





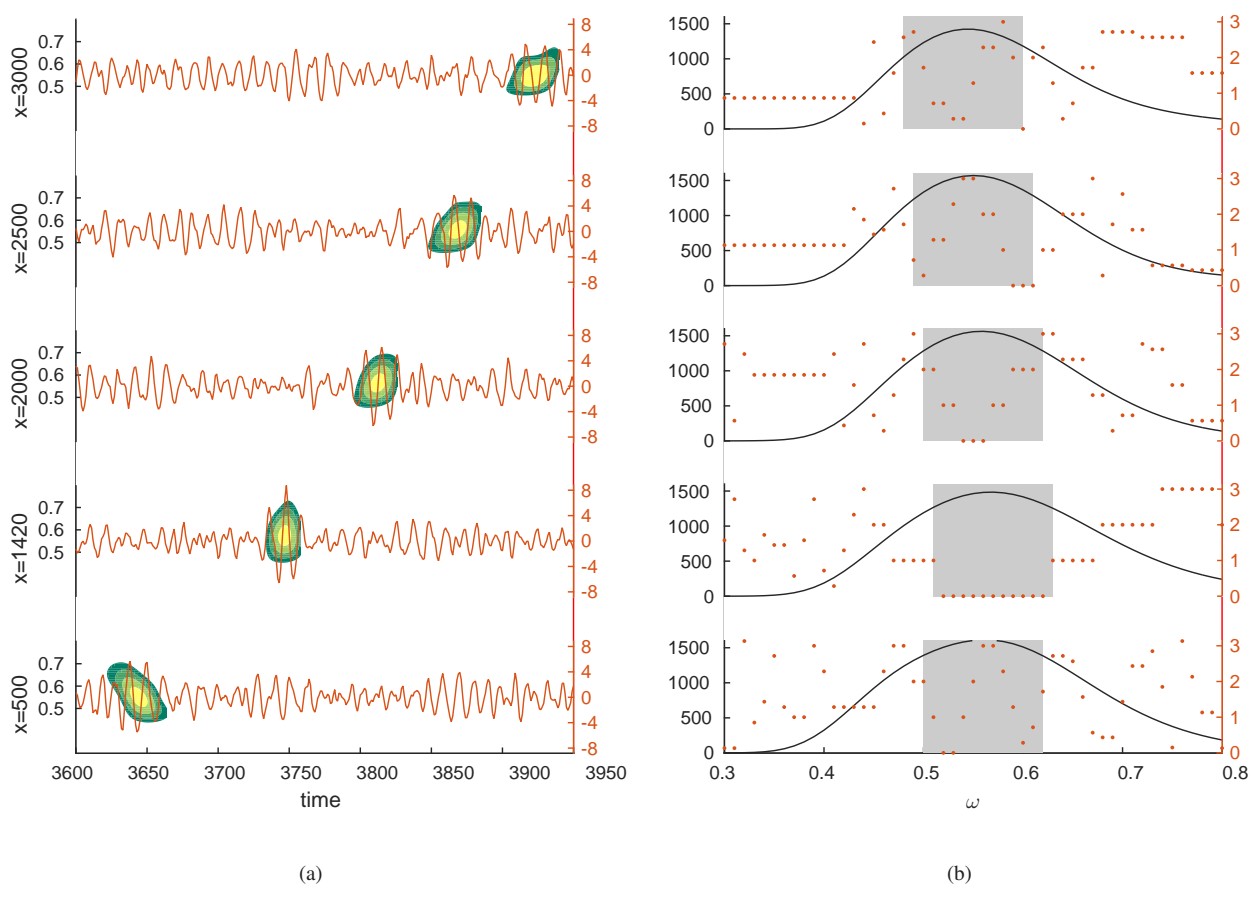

(a)  (b)

**Figure 11.** Case W100. (a) Time signals at various positions of the evolution of the critical group event with the filled contour plot of wavelet spectra. (b) The corresponding time-averaged wavelet spectra (solid line) and the time spreading (dotted line). Observe that at $x = 1420$m the time spreading is zero in the shaded area. The shaded areas show the chosen frequency interval of the most energy carrying modes.





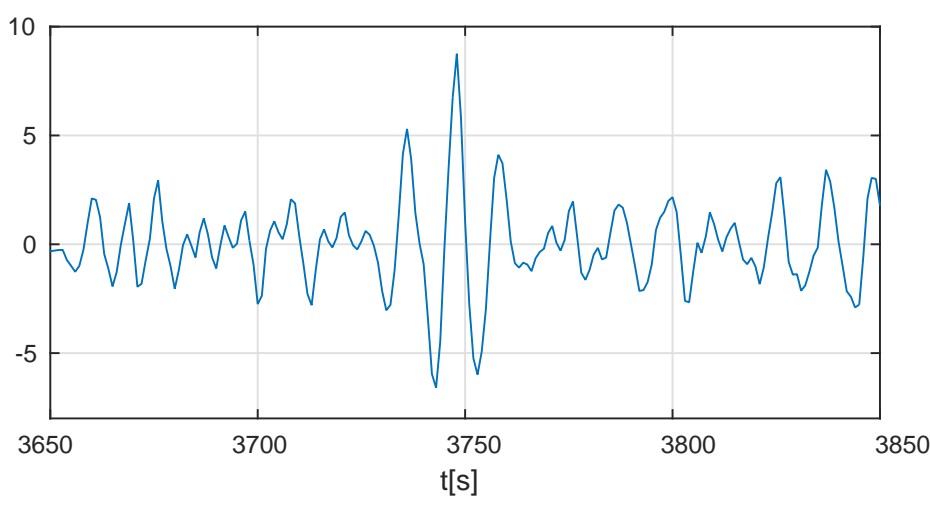

**Figure 12.** Case W100. Zoomed version of the freak wave; the crest height is 1.35 and the wave height is 2.37 times the significant wave height.

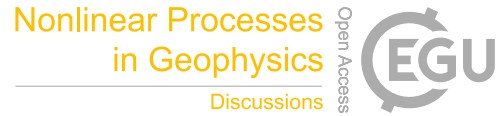


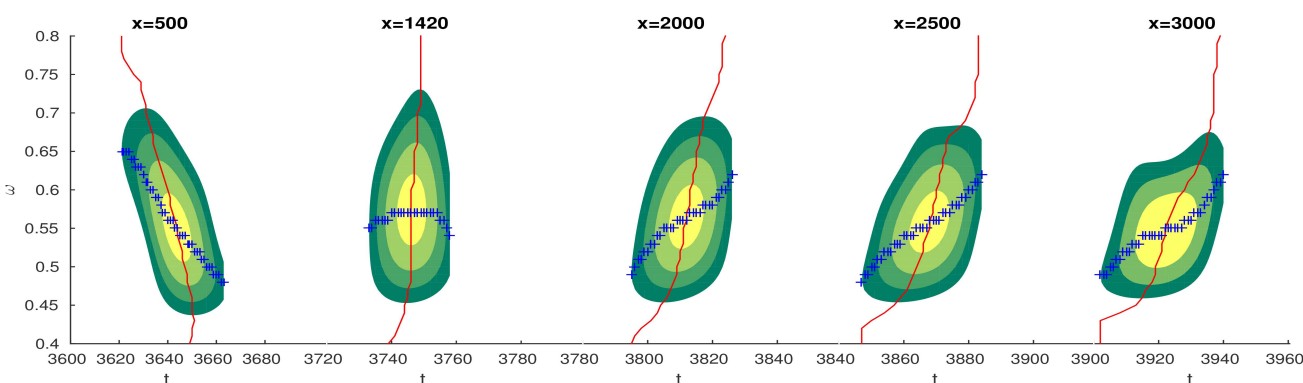

**Figure 13.** Case W100. A filled contour plot of the energy distribution of the critical group event at various positions. At each position the red solid lines show the time of maximal energy at each wave frequency. The '++' lines show the wave frequency as function of time. Both are estimated by the most energetic waves in time and in frequency respectively. Both lines show a decreasing frequency before the freak wave and an increasing frequency after the freak wave, while the freak wave occurs at $x = 1420$m.





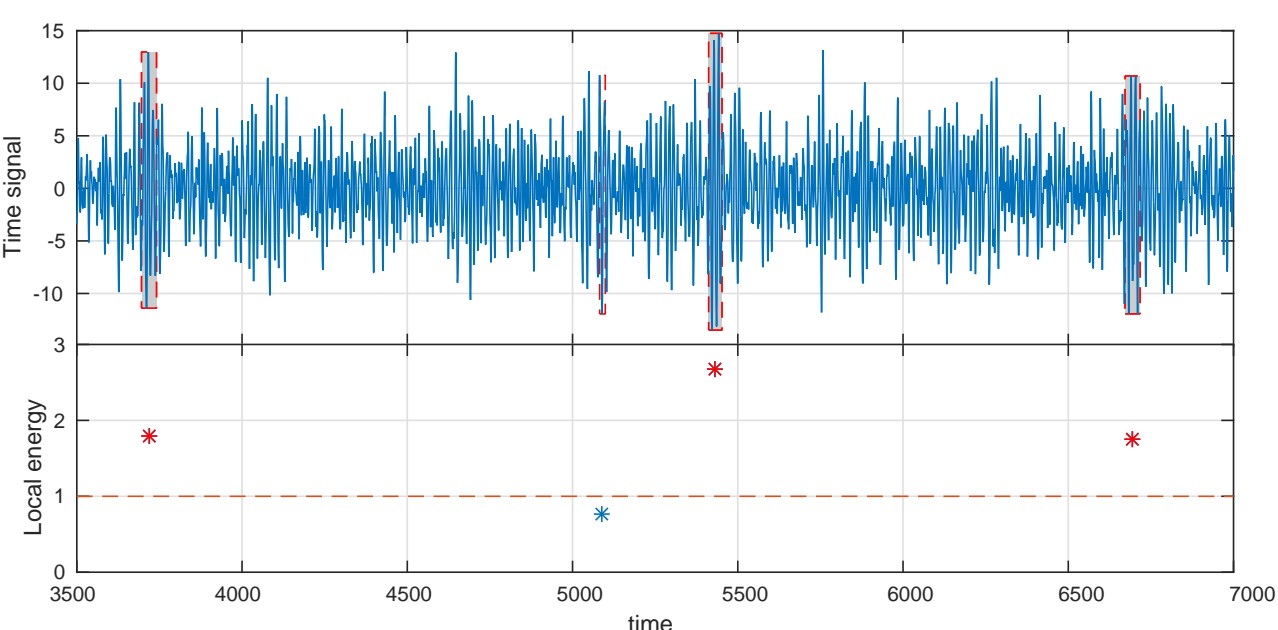

**Figure 14.** Case TS10000. Initial time signal in the interval $t \in [3500, 7000]$. The critical group events are shown in the shaded areas of the upper plot. The lower plot presents the amount of local energy signal of the recognized group events compared to the local energy threshold (dashed line). The local energy signal of the critical group events are above the threshold.





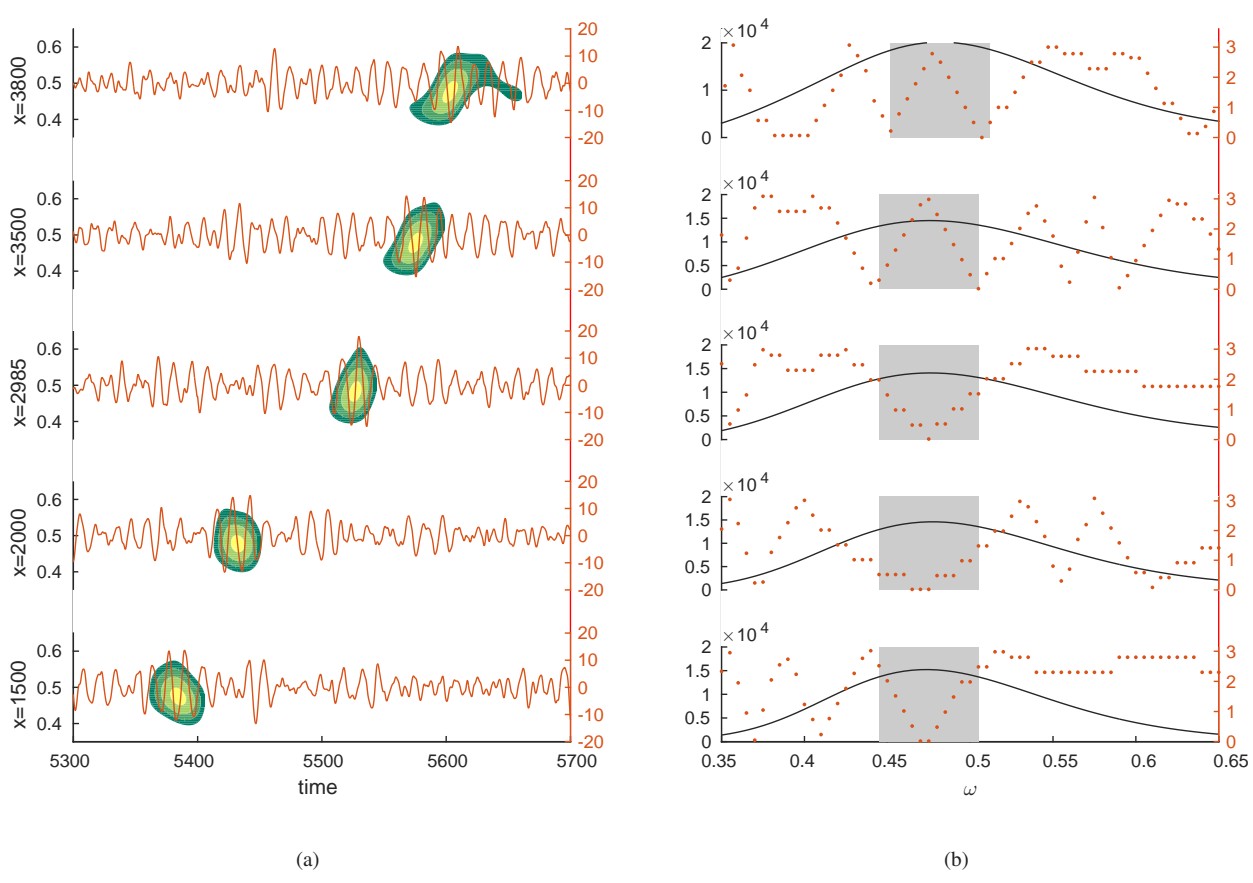

(a)  (b)

**Figure 15.** Case TS10000. (a) Time signals at various positions of the evolution of the critical group event with the filled contour plot of wavelet spectra. (b) The corresponding time-averaged wavelet spectra (solid line) and the time spreading (dotted line). The shaded areas show the chosen frequency interval of the most energy carrying modes.





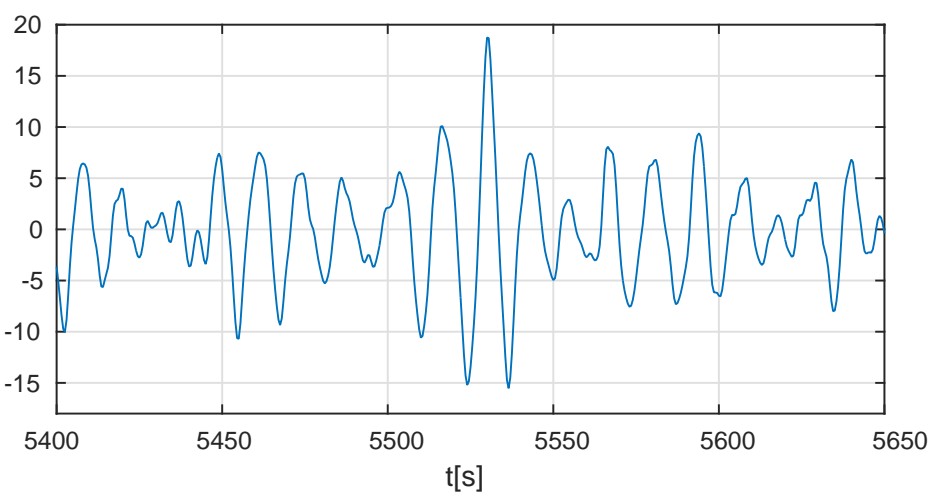

**Figure 16.** Case TS10000. Zoomed version of the freak wave; the crest height is 1.22 and the wave height is 2.23 times the significant wave height.



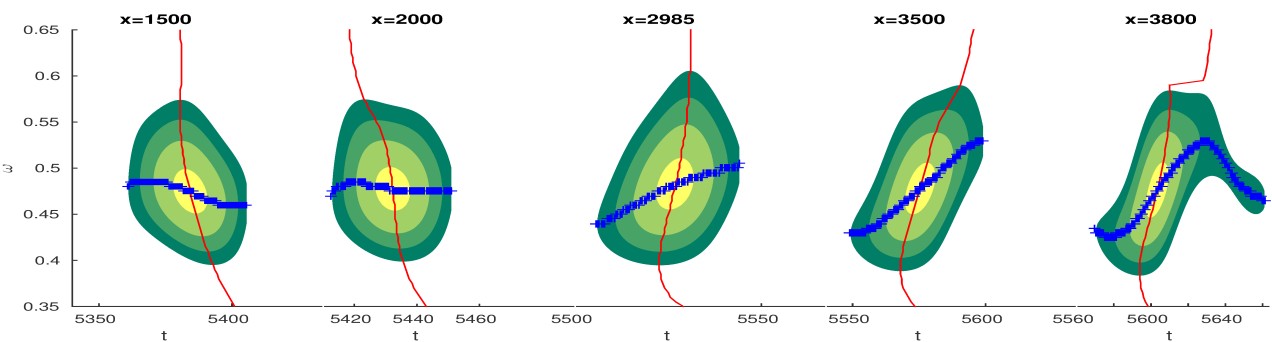

**Figure 17.** Case TS10000. A filled contour plot of the energy distribution of the critical group event at various positions. At each position the red solid lines show the time of maximal energy at each wave frequency. The '++' lines show the wave frequency as function of time. Both are estimated by the most energetic waves in time and in frequency respectively.





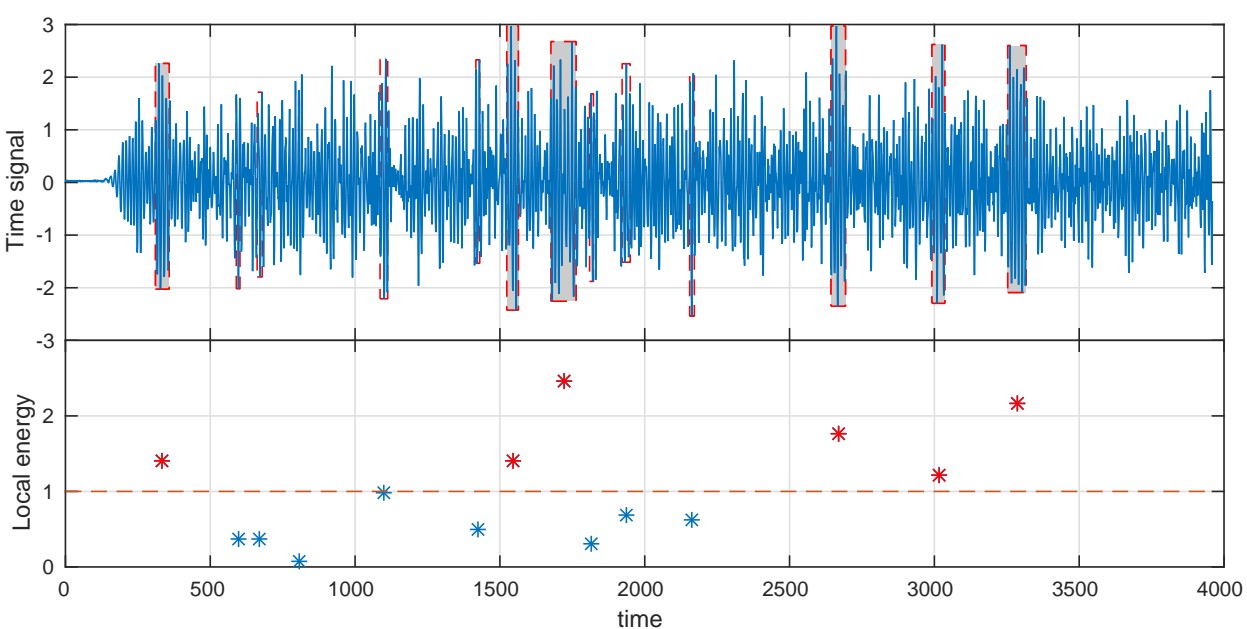

**Figure 18.** Case IW12. The upper plot shows the influx signal. Four critical group events are shown in the shaded areas. The lower plot shows the local energy signal of group events compared to the local energy threshold (dashed-line). The local energy signal of the critical group events are above the threshold.



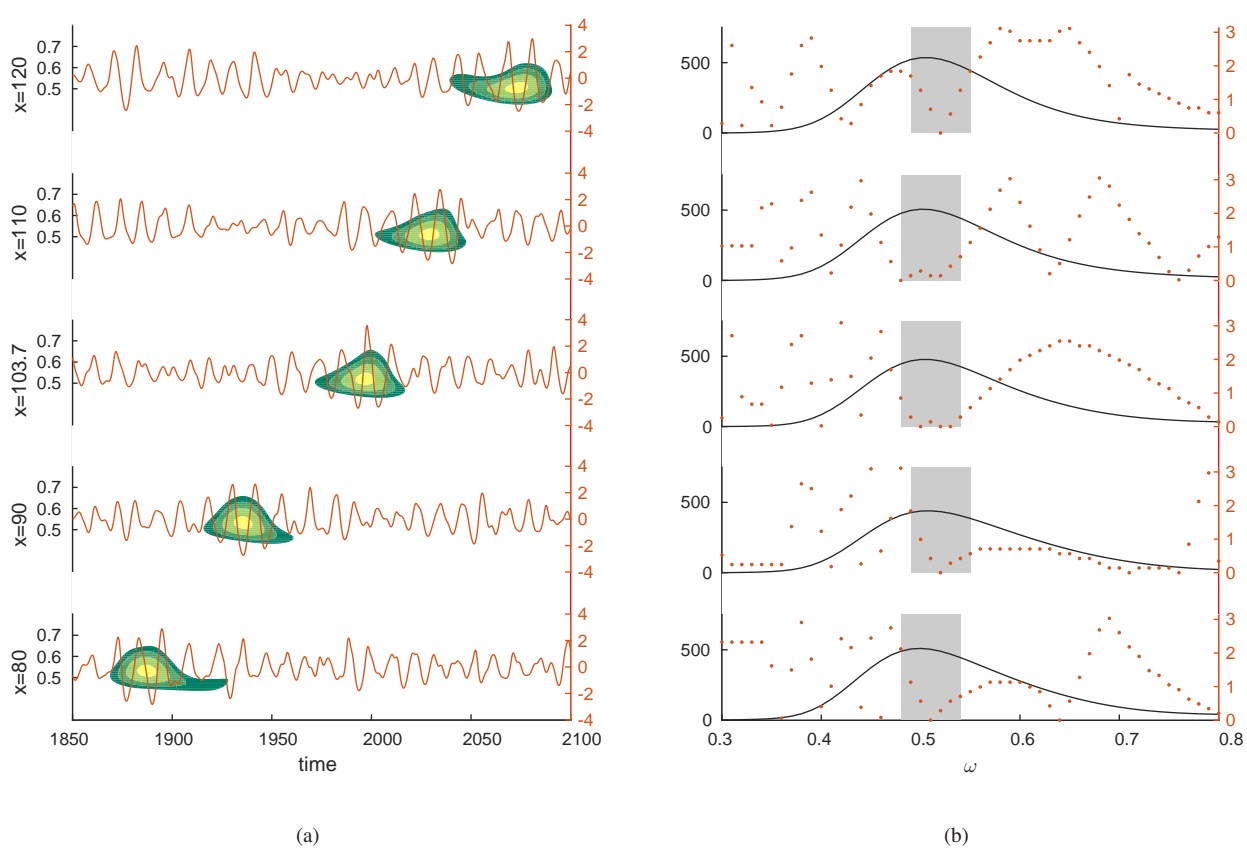

(a)                                              (b)

**Figure 19.** Case IW12. (a) Time signals at various positions of the evolution of the critical group event with the filled contour plot of wavelet spectra. (b) The corresponding time-averaged wavelet spectra (solid line) and the time spreading (dotted line). The shaded areas show the chosen frequency interval of the most energy carrying modes.



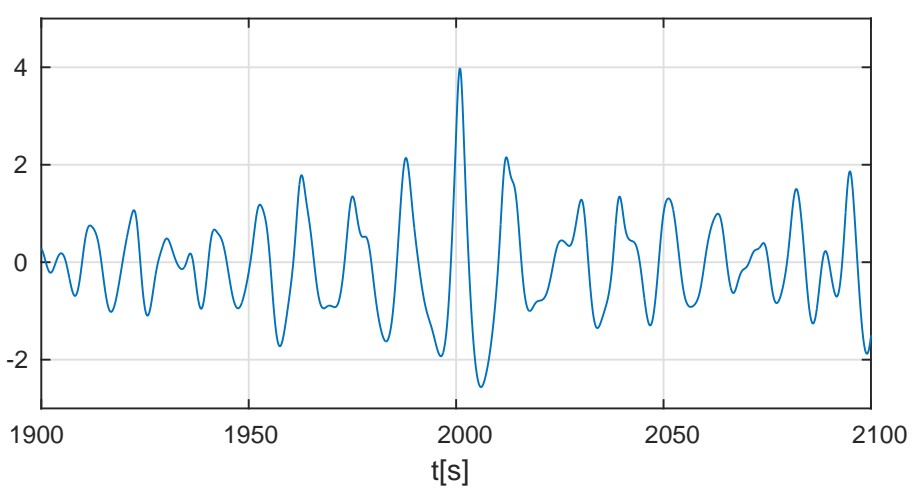

**Figure 20.** Case IW12. Zoomed version of the freak wave; the crest height is 1.31 and the wave height is 2.15 times the significant wave height.





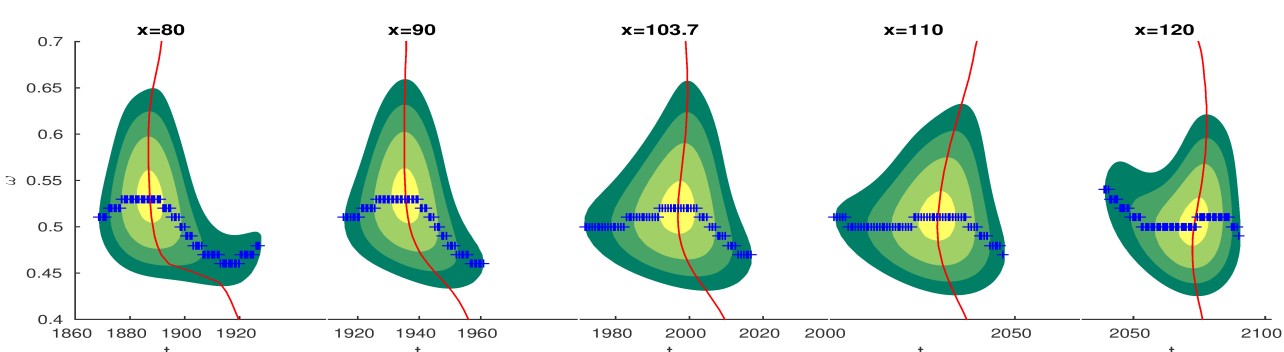

**Figure 21.** Case IW12. A filled contour plot of the energy distribution of the group event at various positions. At each position the red solid lines show the time of maximal energy at each wave frequency. The '++' lines show the wave frequency as function of time. Both are estimated by the most energetic waves in time and in frequency respectively.