# Peer review of "Localized Coherence of Freak Waves"

_Nonlinear Processes in Geophysics, 2016_

## Referee Comment (RC1) · Anonymous Referee #1 · 30 May 2016

This paper is devoted to the practical use of nonlinear-dispersive mechanism of wave focusing for the prediction of freak wave formation. The idea of nonlinear-dispersive mechanism based on coherence of frequency-modulated components in the wave field has become popular nowadays again and allows forecasting the freak wave formation for weakly nonlinear water waves (low and moderate Benjamin-Feir index) before the event occurs. That is why I support publishing the given paper. I like the simplest formulas derived in the manuscript which can estimate the possible maximal amplitude of freak waves. Meanwhile, the whole text is presented in a difficult to read way as a lot of definitions of new integral characteristics are introduced. Being a theoretician, I do not properly understand their practical role, and it will be better to have a reviewer from the experimentalists. I have thought for a long time how to improve the presentation of main quantitative results, but, in fact, I have no idea how to do this radically. I may suggest only some minor comments:

1. Page 2 line 10. It is incorrect that "Pelinovsky et al. (2011) discussed the scenario of a single freak wave in deep water by dispersive focusing of a unidirectional wave packet in linear theory and showed that the freak wave is originated from an anomalous solitary wave". A freak wave of the solitary-like shape is originated from the wave packet.

2. Equation (3). What is the second argument in eta (0,0)? The first argument is t=0, but what about the second one?

3. The authors use the JONSWAP spectrum. What is the used value of gamma? Is it 2 as on page 12?

4. Page 5 line 10. Value 3.8-E8 should be better replaced by 3.8*10^{-8}.

5. AB equation. It is not familiar for readers and perhaps it should be given in Appendix.

6. In the list of references there are no titles for two papers: 1) Baldock, 2) Kharif et al, JETP Letters. The following reference: Pelinovsky et al Physica D, 2000 appears twice. In the book "Rogue waves in the ocean" the author's name is misspelt: it should be Pelinovsky, but not Phelinovsky.

I would like to recommend publishing this paper with some minor revision but I stress the point that the paper can be improved......

---

## Author Comment (AC1) · 13 Jun 2016

We thank the anonymous referee #1 for the positive and constructive remarks. Just like the referee, we struggled to improve and simplify the presentation of the technical results. In section 2 we tried to illustrate in a simple way the role of coherence by considering the (weak) pseudo-maximal signals. Then, to justify the use of the wavelet transform, we showed how the coherence of phase information could be obtained using the information at a single time, and then how much more precise and meaningful phase information is obtained from wavelet techniques, shown graphically in Figures 5 and 6.

The new integral expressions try to quantify the coherence.

An example of the practical use of the results could be early warning of freak waves (for the time being for long crested waves) from radar images of the sea far away from a ship, a topic that is investigated in ongoing research (see Wijaya et al., 2015 and

[Figure]

Wijaya and Groesen, 2016).

The more technical suggestions and comments of the referee have been dealt with as follows:

1. Page 2 line 10 will be revised as follows:

   "Pelinovsky et al. (2011) discussed a freak wave of the solitary-like shape that is originated from the wave packet and is based on the dispersive focusing of unidirectional wave packets."

2. Based on the pm signal definition in (1), there should be only one argument $(t = 0)$ in the equation (3). We have revised it as $[\eta_{pm}(0)]_\alpha$.

3. No, it is not 2. The $\gamma$ is 1.9 as on page 11. We consider the JONSWAP spectrum in the normal sea condition for the illustration mentioned on page 4 line 13 and Figure 2. We added this note in the caption of Figure 2 as follows:

   "The random signal corresponds to a Jonswap spectrum with Hs = 6.3m and $\gamma = 1.9$."

4. We believe what Anonymous Referee #1 meant is page 7 line 10. We have replaced the value as suggested, $3.8 * 10^{-8}$.

5. We added an Appendix that briefly describes the AB equation. It can be seen in the Supplement file.

6. We have revised the list of references and also fixed the typo in the mentioned reference, as the following:

   Baldock, T. E., Swan, C., and Taylor, P. H.: A laboratory study of surface waves on water, Phil Trans. R. Soc. Lond. A, 354, 649–676, 1996.

Kharif, C., Pelinovsky, E., Talipova, T., and Slunyaev, A.: Focusing of Nonlinear Wave Groups in Deep Water, JETP Letters, 73, 170–175, 2001.

Kharif, C., Pelinovsky, E., and Slunyaev, A.: Rogue waves in the Ocean, Advances in Geophysical and Environmental Mechanics and Mathematics, Springer-Verlag, Berlin Heidelberg, 2009.

**Supplement:**

[revised manuscript text omitted]

---

## Referee Comment (RC2) · Anonymous Referee #2 · 17 Jun 2016

The paper presents an analysis of records of water surface elevation over extended periods of time in order to determine the likelihood of appearance of extremely steep (rogue) waves at certain later instant. To this end, both Fourier and wavelet analysis are used extensively, and quantitative criteria are suggested to select segments of the records that contain locally energetic and coherent waves with a significant potential for evolving into a rogue wave phenomenon. The criteria are based on the complex Morlet wavelet spectra that take into account wave characteristic in the neighborhood of each instant. For reader's convenience, the paper contains a separate section with the basics of the wavelet transform. The approach offered by the authors to study rogue waves is interesting and novel so that the paper can be accepted for publication. I have, however, some comments that should be addressed prior to final acceptance. The general idea of the adopted approach is based on dispersive wave focusing, where relatively slow shorter waves precede faster longer waves, and thus may arrive at some later instant and location instantaneously with the same or similar phase. The validity

of this approach is confirmed by analysis of several signals, part of them recorded in wave tanks experiments, while the others were generated by numerical simulations. The common feature of all those examples is that only unidirectional waves were considered. Prediction of dispersive focusing in the presence of directional spreading that it is always present in real ocean waves, may be much more complicated. I believe the manuscript can be improved notably if an example that contains directional spreading is considered. At the very least, the possible effects of angular spreading on the performance of the suggested algorithm should be discussed. Throughout the manuscript, there is no clear distinction between dimensional and dimensionless values. For example, wavelet theory is apparently presented in the dimensionless form, whereas the experimental results are probably dimensional (judging from the reference to time in seconds in the text); the temporal axis in most graphs is titled as 't' or 'time', no units are given. Similarly, it remains unclear what exactly is meant by 'Time signal' (is it elevation in meters?) or Local energy'. In some figures, there are no axis titles at all (e.g. Fig. 9). In all examples, the wave field characteristics such as peak frequency, spectral shape and significant wave height should be given. In Section 4.3, experiments in MARIN are studied. For some reason, scaling (1:50) is mentioned. The scaling is irrelevant to the goals of the present study, however, the reader becomes confused whether the parameters (at least those mentioned in the manuscript) represent the actual measured values or they are scaled. I fully agree with the 1st reviewer, some clarifications regarding the AB model should be provided. The name of Pelinovsky is misspelled on p. 15, line32.

---

## Author Comment (AC2) · 22 Jun 2016

We thank the referee for his remark that the study is interesting and novel and can be accepted for publication. The referee is correct to note that our approach is based on the dispersive effect to explain at least one form of freak wave appearance. We agree very much with the referee that it would be very interesting to broaden the research from uni-to multi-directional waves. To our present understanding, directional spreading in realistic sea states could very well influence a dispersive focussing effect. We refer to the interesting paper,

Adcock TAA, Taylor PH, Draper S.: Nonlinear dynamics of wave-groups in random seas: unexpected walls of water in the open ocean. Proc. R. Soc. A, 471(2184): 2015.0660, 2015.

Numerical simulations in that paper indicate that nonlinearity, on average, does not increase the amplitude of linear extreme events, but that the shape of the extreme crest is changed. Details of the shape, coherence and local propagation directions in (linear) random seas is however not part of that study. This makes the suggestion of the referee to extend our investigations to multi-directional waves even more understandable and desirable. However, the referee may agree that such investigations go far beyond the scope of the present manuscript and will require substantial time and effort. Our interest for the extension to multi-directional case is clear, and we may have the analytical and numerical means to investigate this topic, but cannot at this moment foresee or speculate when new insights are mature enough for publication.

The more technical comments of the referee have been dealt with as follows:

- In the basic theory such as Fourier and wavelet transformation in Section 2 and 3, the dimensionless form are used. We used the dimension form in the study cases, therefore we add the corresponding units in all results in Section 4.

- To be consistent, we have corrected the temporal axis in all graphs with title 't' and added the unit. The 'Time signal' means the surface elevation in meters. To make it clear, we change the axis figures labelled by 'Time signal' to be $\eta$[m]. We also add axis in some figures with no axis before.

- The wave field characteristics, i.e. the peak period (instead of peak frequency) and the significant wave height are already given in the paragraphs, except in the dispersive focusing waves. Then, we will add the characteristics for the dispersive focusing waves. Additionally, we also add the spectral shape of the influx signal for each example.

- We agree with the referee that the scaling (1:50) in section 4.3 is irrelevant to the goals of the paper and makes a confusing for the readers. Therefore, we will

present the fourth study case without any scaling and adjust the results related to the case with the actual measured dimension in the laboratory experiment.

- The AB model has been added as an appendix.

- The misspelled name of Pelinovsky has been corrected.

All changes and corrections can be seen in the supplement file.

Please also note the supplement to this comment:
http://www.nonlin-processes-geophys-discuss.net/npg-2016-31/npg-2016-31-AC2-supplement.pdf
* * *
[Figure]

**Supplement:**

[revised manuscript text omitted]

---

## Referee Comment (RC3) · Anonymous Referee #2 · 30 Jun 2016

In the revision, the authors took into account appropriately most my comments, with a single exception. In my previous review, I did not necessarily request to broaden the research to multi-directional waves. However, I stressed that it should be clearly stated that only unidirectional waves are considered, and the possible effects of directional spreading are briefly discussed. In their reply to my comment about directional spreading, the authors refer to Adcock et al. (2015), but this paper (or any other publication that deals with that point) remains not mentioned in the revised manuscript.

---

## Author Comment (AC3) · 3 Jul 2016

We apologize to the referee for the fact that we did not make the remarks about the restriction to uni-directional waves more explicit in the revised manuscript. This is now improved by adding the following sentences in the Introduction, including some references, see page 2, starting at line 10, as follows:

"In addition to the references cited above, we will contribute in understanding the process and the origin of freak wave appearance in random wave fields that is mainly based on dispersive effects. In realistic sea states a directional spreading could possibly influence a dispersive focussing effects (Prevosto, 1998). Also Johannessen and Swan (1998) concluded that the introduction of directionality significantly reduces the nonlinearity of wave groups. That nonlinearity gives little or no extra amplitude compared to linear extreme events, but that shape of the extreme crest is

changed was also observed by Adcock et al. (2015). In this paper, we will not take directional spreading into account, but will restrict to long-crested, uni-directional waves."

Johannessen, T. and Swan, C.: Extreme multi-direction waves, Coastal Engineering Proceeding, 1(26), 1998.

Adcock, T. A. A., Taylor, P. H., and Draper, S.: Nonlinear dynamics of wave-groups in random seas: unexpected walls of water in the open ocean, Proc. R. Soc. A, 471, 2015.

Prevosto, M.: Effect of Directional Spreading and Spectral Bandwidth on the Nonlinearity of the Irregular Waves, in: Proceedings of the Eighth International Offshore and Polar Engineering Conference, pp. 119-123, 1998.

Please also note the supplement to this comment:
http://www.nonlin-processes-geophys-discuss.net/npg-2016-31/npg-2016-31-AC3-supplement.pdf
* * *
[Figure]

**Supplement:**

[revised manuscript text omitted]